# Glacial-interglacial Circumpolar Deep Water temperatures during the last 800,000 years: estimates from a synthesis of bottom water temperature reconstructions

David M. Chandler[1] and Petra M. Langebroek[1]

[1]NORCE Norwegian Research Centre, Bjerknes Centre for Climate Research, Bergen, Norway

**Correspondence:** David Chandler (dcha@norcereserch.no)

**Abstract.** Future climate and sea-level projections depend sensitively on the response of the Antarctic Ice Sheet to ocean-driven melting and the resulting freshwater fluxes into the Southern Ocean. Circumpolar Deep Water (CDW) transport across the Antarctic continental shelf, and into cavities beneath ice shelves, is increasingly recognised as a crucial heat source for ice shelf melt. Quantifying past changes in the temperature of CDW is therefore of great benefit for modelling ice sheet response to past warm climates, for validating paleoclimate models, and for putting recent and projected changes in CDW temperature into context. Here we compile the available bottom water temperature reconstructions representative of CDW over the past 800 kyr. Estimated interglacial warming reached anomalies of +0.6±0.4°C (MIS 11) and +0.5±0.5°C (MIS 5) relative to present. Glacial cooling typically reached anomalies of ca. -1.5 to -2°C, therefore maintaining positive thermal forcing for ice shelf melt even during glacials in the Amundsen Sea region of West Antarctica. Despite high variance amongst a small number of records, and poor (4 kyr) temporal resolution, we find persistent and close relationships between our estimated CDW temperature and Southern Ocean sea-surface temperature, Antarctic surface air temperature, and global deep water temperature reconstructions at glacial cycle time scales. Given the important role that CDW plays in connecting the world's three main ocean basins, and in driving Antarctic Ice Sheet mass loss, additional temperature reconstructions targeting CDW are urgently needed to increase temporal and spatial resolution and to decrease uncertainty in past CDW temperatures – whether for use as a boundary condition, model validation or to understand past oceanographic changes.

## 1 Introduction

In its position at the southern end of the Atlantic meridional overturning circulation (AMOC), the Southern Ocean plays a major role in the Earth's climate. Interactions between the Southern Ocean and Antarctic Ice Sheet drive changes in ice discharge and ocean circulation, which in turn have global-scale impacts on sea-level and climate (Bronselaer et al., 2018; Rintoul, 2018; Mackie et al., 2020b; Noble et al., 2020). Complex feedbacks between grounded ice dynamics, ice shelves and ocean circulation are either poorly understood and/or difficult to capture at adequate resolution in numerical simulations (Bronselaer et al., 2018; Edwards et al., 2019; Fox-Kemper et al., 2021; Bamber et al., 2022; Siahaan et al., 2022). As a result, the contribution of Antarctica to future sea-level and climate change remains very uncertain (Fox-Kemper et al., 2021; Bamber et al., 2022).

To help inform future climate and sea-level projections, we can reconstruct or simulate past climates, and their corresponding ocean and ice sheet responses, for previous periods when the Earth was warmer than present. These experiments complement future projections as they represent climate and ice sheet states outside of those for which direct observations are available (Tzedakis et al., 2009; Gilford et al., 2020; DeConto et al., 2021). As the most recent warmer period in geological history, the last interglacial (LIG) has seen widespread interest as an analogue for future warming, from the perspectives of both paleoclimate (Mercer, 1984; Bakker et al., 2014; Arias et al., 2021; Otto-Bliesner et al., 2021; Zhang et al., 2023) and the resulting ice sheet response (Mercer, 1978; Scherer, 1991; Goelzer et al., 2016; DeConto et al., 2021; Golledge et al., 2021). This interest likely reflects the relatively greater availability and/or resolution of LIG paleoenvironmental reconstructions, compared to those for older warm periods. However, there have been many Quaternary interglacials before the LIG, some a little warmer or cooler than the present (Tzedakis et al., 2009; Yin and Berger, 2015; Past Interglacials Working Group of PAGES, 2016). Climate and ice sheet simulations through several glacial-interglacial cycles build a much more complete picture of feedbacks, instabilities and tipping points in the Earth system. Of particular interest is the period after the mid-Pleistocene transition (MPT, at ca. 1250-700 ka: Clark et al., 2006; Legrain et al., 2023), and particularly marine isotope stages (MIS) 11, 9, 7 and 5, since these best represent the 100-kyr glacial cycles of our pre-industrial interglacial period. However, we should note that the moderate (SSP2-4.5) and high (SSP5-8.5) IPCC warming scenarios could take us beyond the global surface warming magnitudes reconstructed in any Quaternary interglacial (IPCC 6th Assessment Report, Technical Summary: Arias et al., 2021).

Southern Ocean sea-surface temperature (SST) and sea-ice extent during the LIG and preceding glacial (the penultimate glacial maximum, PGM) have been the focus of several recent synthesis studies (Hoffman et al., 2017; Turney et al., 2020; Chandler and Langebroek, 2021b; Chadwick et al., 2022). Climatic changes in the Southern Ocean extend far deeper than the surface waters: the region's role in the AMOC (Talley, 2013; Buckley and Marshall, 2016; Rintoul, 2018; Carter et al., 2022), deep ocean heat storage (Gjermundsen et al., 2021), and in delivering warm water *at depth* to cavities beneath Antarctic ice shelves (Walker et al., 2007; Wåhlin et al., 2010; Herraiz-Borreguero et al., 2015; Silvano et al., 2017), means that changes in the temperature of deep water masses are a crucial consideration in both climate and ice sheet simulations. Upwelling circumpolar deep water (CDW) around Antarctica is particularly important, owing to the sensitivity of key Antarctic Ice Sheet sectors to sub-surface ocean warming (Walker et al., 2007; Herraiz-Borreguero et al., 2015; Silvano et al., 2017; Reese et al., 2018; Noble et al., 2020; van Wijk et al., 2022).

Despite the apparent importance of deep water masses, there is still a strong reliance on surface temperature reconstructions for evaluating climate models (e.g., Otto-Bliesner et al., 2021; Purich and England, 2021). The bias presumably reflects the availability of proxy reconstructions, but at the same time there is no guarantee that models selected on the basis of their match to surface conditions will show similar skill in simulating deeper levels. This is particularly the case in the Southern Ocean, where processes that are not yet well represented in CMIP models can drive subsurface temperature changes in directions opposite to those at the surface (Rintoul, 2018; Mackie et al., 2020b; Bronselaer et al., 2018). Indeed, significant difficulties in simulating Southern Ocean warming, and the Antarctic Ice Sheet response to warming, are the strong ice - ocean - atmosphere interactions at sub-grid scales (Heywood et al., 2014; Hewitt et al., 2020; Mackie et al., 2020a; Purich and England, 2021).

However, the high computational cost of fully-coupled models limits experiments to short (decadal) time periods (e.g., Kreuzer et al., 2021; Pelletier et al., 2022; Siahaan et al., 2022).

Motivated by the need for CDW temperature reconstructions to constrain the ocean temperature boundary condition in stand-alone (uncoupled) Antarctic Ice Sheet simulations, this paper synthesises the sparse proxy data that are available to estimate changes in CDW temperature over the last 800,000 years. This time span was selected for five reasons: (i) it covers the period for which there are sufficient data to establish a meaningful synthesis at a reasonable temporal resolution; (ii) it covers the 100 kyr glacial cycles most relevant to our present climate state, albeit before the onset of anthropogenic influences; (iii) inclusion of colder interglacials prior to MIS 11 can provide a more detailed picture of Earth system response to warming; (iv) it matches the duration of the longest Antarctic ice core record (EPICA Dome C: Jouzel et al. 2007); and (v) deep water temperature proxy records derived from oxygen isotopes (the main data source here - see methods) are considered less reliable prior to the MPT (Bates et al., 2014).

The 4-kyr temporal resolution of our synthesis is currently too coarse for practical application directly as a boundary condition for Antarctic Ice Sheet models. Furthermore, we have not reconstructed changes in the rate of transport of CDW towards ice shelf cavities, which is currently only possible in the Holocene (e.g. Hillenbrand et al., 2017; King et al., 2018; Xu et al., 2021). Hence, the scope of this paper is to provide a best estimate of CDW temperature changes at these time scales, without going further to calculate potential changes in Antarctic ice shelf melting. However, the synthesis can be used to validate alternative CDW temperature estimates that have recently been employed in ice sheet models without independent validation (e.g. Quiquet et al., 2018; Tigchelaar et al., 2018; Sutter et al., 2019; Albrecht et al., 2020); it can also help in evaluating transient or time-slice climate model output.

## 2   Oceanographic setting

This section provides a brief summary of how CDW fits within the complex Southern Ocean circulation, included here as context for the temperature reconstructions and site selection. The cited studies (particularly Talley, 2013, and Carter et al., 2022) provide much more detail.

CDW comprises the relatively warm water mass forming the bulk of the water within the Antarctic Circumpolar Current (ACC) (Pardo et al., 2012). North of the Southern Polar Front, CDW lies at intermediate depths between the underlying Antarctic Bottom Water (AABW) and the overlying Antarctic intermediate water (AAIW) (Fig. 1). South of the Southern Polar Front, upwelling brings CDW towards the surface, driven partly by diverging Ekman transport beneath the mid-latitude westerlies and polar easterlies (Pardo et al., 2012; Talley, 2013; Tamsitt et al., 2021; Carter et al., 2022), and partly by bathymetry (Tamsitt et al., 2017). CDW can be further classified into its lower (LCDW) and upper (UCDW) components, of which the LCDW is the more significant for ice shelf basal melt in Antarctica (Orsi et al., 1995; Jacobs et al., 1996; Adusumilli et al., 2020; Carter et al., 2022).

CDW is not a water mass formed directly by ocean surface processes, but is instead a mixture of deep water masses entering the ACC from each of the three main ocean basins. In the Atlantic sector, southwards-flowing upper NADW (primarily sourced

in the Labrador Sea) upwells to join the UCDW, subsequently returning northwards as AAIW in the upper leg of the AMOC (Fig. 1). Meanwhile, southwards-flowing lower NADW (sourced primarily from ice-ocean interactions in the Greenland and Norwegian Seas) upwells to join the LCDW (Smethie et al., 2000; Talley, 2013; Tamsitt et al., 2017). In the Indian basin, AABW and an eastbound branch of NADW spread northwards and are gradually mixed diffusively with overlying deep water, before returning southwards as Indian deep water and joining the UCDW (Talley, 2013). In the Pacific sector, this same diffusive process returns northwards-spreading AABW and LCDW southwards as Pacific deep water, again most likely joining the UCDW (Kawabe and Fujio, 2010; Talley, 2013; Biddle et al., 2017; Assmann et al., 2019). Finally, on its path around the ACC, LCDW is cooled by mixing with other Antarctic water masses (e.g., Weddell Sea deep water), particularly in the Scotia Sea (Naveira Garabato et al., 2002; Carter et al., 2022).

Although the ACC flows (on average) mainly eastwards, LCDW within the ACC is transported southwards towards the Antarctic continental shelf by a range of processes including eddies, internal waves, topographic influences, and Ekman transport (Stewart and Thompson, 2015; Thompson et al., 2018; Tamsitt et al., 2017, 2021; Darelius et al., 2023). Mixing with colder local surface and shelf water masses further transforms LCDW temperature and salinity if it crosses the shelf (MacAyeal, 1984; Pardo et al., 2012; Petty et al., 2013). This cross-shelf transport of modified CDW enters the cavities beneath some Antarctic ice shelves, including those in the Amundsen Sea Embayment, and those draining the Aurora basin of East Antarctica (Walker et al., 2007; Wåhlin et al., 2010; Silvano et al., 2017; van Wijk et al., 2022). Even the modified CDW has sufficient thermal forcing to cause rapid ice shelf basal melting (exceeding 10 m yr$^{-1}$: Adumusilli et al., 2020) near Antarctic grounding lines. Modified CDW does not currently enter cavities beneath 'cold' ice shelves such as Filchner-Ronne or Ross, except in very localised regions (Darelius et al., 2023). Instead, beneath these 'cold' ice shelves, slower rates of basal melt are driven mainly by high salinity shelf water (HSSW) originating from sea-ice processes (brine rejection) (MacAyeal, 1984; Petty et al., 2013). It is this ocean-driven melting of ice shelves, and its resulting freshwater release, that is a crucial process in Antarctic Ice Sheet models and climate models over a broad range of time scales.

In our synthesis of CDW temperatures we select sites currently bathed in LCDW (Section 3.3). There is no guarantee that the same water masses have persisted at these sites – particularly during climates increasingly different from present. This possibility is discussed further in Section 5.3.

# 3 Methods

## 3.1 Bottom water temperature reconstructions from proxies

CDW paleotemperature estimates are dependent on bottom water temperature reconstructions, which in the Southern Ocean are based on just two proxies. Both use benthic foraminiferal calcite: these are the Mg/Ca ratio, and oxygen isotopic composition ($\delta^{18}O_b$).

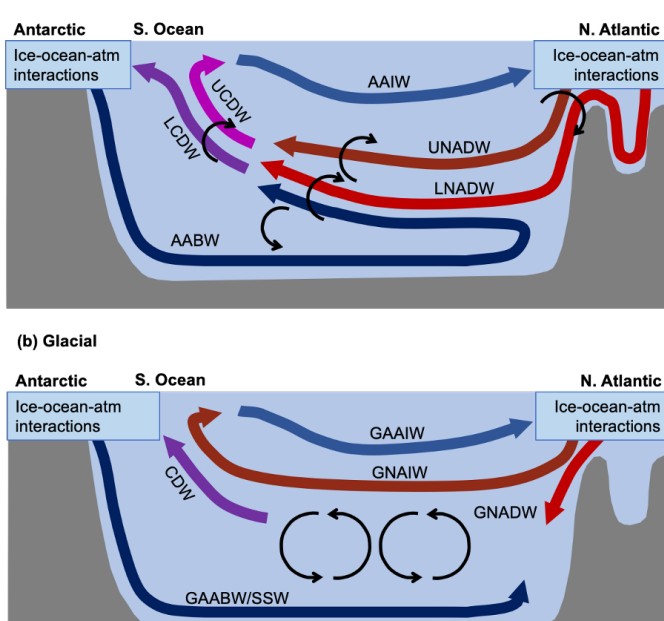

**Figure 1.** Schematic diagrams showing how the Atlantic meridional overturning circulation (AMOC) contributes North Atlantic deep water (NADW) to circumpolar deep water (CDW) during present-day (interglacial) and reconstructed glacial climate states. Both schematics highlight the main circulation features in slightly different versions by Ferrari et al. (2014), Howe et al. (2016), and Matsumoto (2017). Note that we only show the meridional (upwelling) component of CDW, whereas its dominant velocity component in reality is eastwards in the Antarctic Circumpolar Current. In both climate states, Antarctic bottom water (AABW) reaching the North Atlantic eventually returns south within a lower cell to mix with water masses upwelling in the Southern Ocean (NADW, CDW, or their glacial equivalents). The properties of upwelling CDW should then reflect contributions from southbound upper NADW or glacial North Atlantic intermediate water (GNAIW) in the upper overturning cell as well as lower NADW and returning AABW in the lower cell. Talley (2013) provides 3D schematics showing the AMOC within the global ocean circulation. See Sections 2 and 5.3 for more details.

### 3.1.1 Mg/Ca ratio

Benthic foraminifera incorporate several trace metals into their calcite shells, including magnesium. The shell Mg/Ca ratio
depends partly on water temperature-dependent thermodynamic and physiological processes during calcification (Chave, 1954; Izuka, 1988; Kunioka et al., 2006), enabling its use as a proxy for seawater paleotemperature. Mg/Ca has been particularly valuable for SST reconstructions in the Southern Ocean (Chandler and Langebroek, 2021a, and references therein), but is unfortunately much less widely used for bottom water reconstructions in this region (Elderfield et al., 2012; Hasenfratz et al., 2019). Calibrations to reconstruct water temperature from benthic foraminiferal Mg/Ca have been developed both for deep
ocean environments (Elderfield et al., 2006; Hasenfratz et al., 2019; Stirpe et al., 2021) and for the Antarctic continental shelf

(Rathbun and De Deckker, 1997; Mawbey et al., 2020), although there are as yet no pre-Holocene sedimentary Mg/Ca records from the latter.

Besides temperature, Mg/Ca is influenced by important non-thermal factors, to varying extents in different species and environments, as well as by laboratory analytical procedures (Elderfield et al., 2006; Yu et al., 2007). Perhaps most importantly in the Southern Ocean is the seawater carbonate chemistry (carbonate ion saturation: $\Delta[\mathrm{CO}_3^{2-}]$), which is considered a secondary control on Mg/Ca (especially at temperatures below $\sim 4°$C) in several foraminifera species (Raitzsch et al., 2008; Elderfield et al., 2006; Tisserand et al., 2013; Stirpe et al., 2021) or possibly a primary control in *Cibicidoides wuellerstorfi* over glacial-interglacial cycles (Raitzsch et al., 2008; Yu and Elderfield, 2008). Corrections for this influence are feasible (e.g. Healey et al., 2008) but only if $\Delta[\mathrm{CO}_3^{2-}]$ can also be reconstructed (which can be done in principle from B/Ca: Yu et al., 2008). Infaunal species which live *within* the surface sediments may be less influenced by $\Delta[\mathrm{CO}_3^{2-}]$ than epifaunal species which live *on* the sediment surface (Elderfield et al., 2006, 2010). The two records included in this synthesis use infaunal species *Melonis pompiloides* and *Uvigerina spp.*. Neither record has been corrected for $\Delta[\mathrm{CO}_3^{2-}]$, but this correction could be considered in future studies if sufficient data are available.

Alternative biogenic calcite trace-metal proxies may be less sensitive to carbonate saturation: these include foraminiferal calcite Li/Mg (Bryan and Marchitto, 2008; Chen et al., 2023); or ostracod calcite Mg/Ca (Farmer et al., 2012). Neither has yet been used as a temperature proxy in the Southern Ocean, but the latter has been used to reconstruct NADW temperature changes in the North Atlantic since MIS 7 (Cronin et al., 2000).

Another influence of carbonate chemistry is post-depositional calcite dissolution. Firstly, dissolution hinders sampling by reducing the amount of material available in deep ocean sites (see Section 3.2 below). Secondly, dissolution can bias Mg/Ca paleothermometry – either if Mg is not distributed evenly through individual shells and only the outer parts of shells are dissolved, and/or if Mg-rich calcite is dissolved preferentially (Hintz et al., 2006; Kunioka et al., 2006). Dissolution begins below the carbonate saturation horizon (about 3100 m in the Southern Ocean: Bostock et al., 2011; Jones et al., 2021) and increases sharply below the lysocline at ca. 4000 m (Williams et al., 1985; Hayward et al., 2001; Bostock et al., 2011). Depths of the two sites with Mg/Ca data in our study are 2807 m (ODP 1094) and 3290 m (ODP 1123), suggesting minimal influence of dissolution under modern conditions. However, the lysocline may have been shallower during glacial climates (Howard and Prell, 1994).

Finally, changes in ambient seawater Mg/Ca ratio are potentially important at Myr timescales (Ries, 2010). Seawater Mg/Ca has increased by only $\sim$0.1 mol/mol/Myr during the last 20 Myr, towards its modern ratio of $\sim$5 mol/mol (Coggon et al., 2010; Evans and Müller, 2012), so that changes in seawater Mg/Ca are not considered important for the purposes of this study.

### 3.1.2 Foraminiferal calcite $\delta^{18}\mathrm{O}_b$

Benthic foraminiferal calcite $\delta^{18}\mathrm{O}_b$ depends on both the temperature $T_{sw}$ and $\delta^{18}\mathrm{O}_{sw}$ of ambient seawater (Urey et al., 1951; Emiliani, 1955; Shackleton, 1967). At glacial cycle timescales, $\delta^{18}\mathrm{O}_{sw}$ varies primarily with global ice volume, as fractionation of $^{16}\mathrm{O}$ and $^{18}\mathrm{O}$ during evaporation and precipitation causes $^{16}\mathrm{O}$ to become preferentially locked up in ice sheets. Consequently, seawater $\delta^{18}\mathrm{O}_{sw}$ becomes more positive as ice volume increases. Local hydrographic changes may also influence $\delta^{18}\mathrm{O}_{sw}$, but

at glacial cycle time scales we follow Siddall et al. (2010) and Bates et al. (2014) who neglect this additional influence. We justify this assumption on the basis that the amplitude of Southern Ocean variability in $\delta^{18}O_{sw}$ through glacial cycles is similar to that of the global average (see estimates of $\Delta\delta^{18}O_{swLGM}$ later in this section). However, the extent to which ice-ocean interactions close to Antarctica might invalidate this assumption – whether at glacial cycle or shorter time scales – will need addressing in future with high-resolution coupled ice sheet - ocean modelling, particularly if future CDW temperature reconstructions use core sites further south than ours. Hence, for now we assume $\delta^{18}O_b$ contains only ambient temperature and global ice volume signals, i.e.,

$$\delta^{18}O_b = \delta^{18}O_T + \delta^{18}O_{ice} \tag{1}$$

Separating the temperature and seawater contributions to $\delta^{18}O_b$ requires firstly estimates of global sea-level, secondly a suitable scaling to convert changes in sea level to changes in $\delta^{18}O_{ice}$, and finally a suitable paleotemperature equation linking benthic calcite $\delta^{18}O_b$, $T_{sw}$, and $\delta^{18}O_{ice}$ (Chappell and Shackleton, 1986; Waelbroeck et al., 2002; Siddall et al., 2010; Bates et al., 2014). These three steps are described below.

Past sea-level estimates at key times, and their likely uncertainty ranges, were estimated using published records derived from multiple sources of evidence including fossil corals, oxygen isotopic analysis of marine sediments, GIA modelling, and cave speleothems (Table 1 and references therein).

The scaling from $\delta^{18}O_b$ to sea-level $S$ is accomplished using site-specific transfer functions (Siddall et al., 2010; Bates et al., 2014). The transfer functions are established using "calibration windows" – typically full glacial or interglacial conditions – for which both sea-level and $\delta^{18}O_b$ have been respectively reconstructed or measured. These functions are then used to convert the $\delta^{18}O_b$ time series in each sediment core to a site-specific sea-level time series $S$. Siddall et al. (2010) and Bates et al. (2014) used linear piece-wise transfer functions relating $S$ to $\delta^{18}O_b$ for sea-level markers in MIS 1 to 5, 7 and the mid-Pliocene (Table 1). However, because of uncertainties in the sea-level estimates and noise in the $\delta^{18}O_b$ records, linear piece-wise functions are susceptible to very variable gradients (which sometimes reverse) between neighbouring points (Fig. 2), that lack a clear physical basis. This is increasingly a problem as more markers are added, particularly when they have similar $\delta^{18}O_b$. Instead, we here develop their approach by (i) updating existing and adding additional sea-level markers that have become available since their studies, as noted above (Table 1); and (ii) establishing second- or third-order polynomial transfer functions for each site instead of linear piece-wise functions. These changes enable a greater sampling of the ($\delta^{18}O_b$, $S$) space so there is less influence of individual (and typically uncertain) events, while the smoothly varying relationship between $S$ and $\delta^{18}O_b$ is more physically plausible.

Since the sea-level markers have varying estimates of uncertainties, we use weighted least-squares regression to calculate the polynomial coefficients ($\beta_i$), with weights equal to the inverse square of the respective sea-level uncertainty ranges in Table 1. Specifically, if we have $N$ pairs of sea-level and $\delta^{18}O_b$, at events $i = 1..N$, the transfer function is

$$S = \beta_0 + \beta_1(\delta^{18}O_b) + \beta_2(\delta^{18}O_b)^2 + \beta_3(\delta^{18}O_b)^3 \tag{2}$$

with the coefficients calculated using

$$\beta = (\mathbf{X^T W X})^{-1} \mathbf{X^T W s} \tag{3}$$

Here $\mathbf{s} = s_i$ is a vector of sea-levels at each event $i = 1..N$; $\mathbf{W}$ is an $N \times N$ matrix with weights $w_i$ on the diagonal and zeros elsewhere (i.e., $W_{i=j} = w_i$, $W_{i \neq j} = 0$); and $\mathbf{X}$ has elements $X_{ij} = \delta_i^k$ where $\delta_i$ is the $\delta^{18}O_b$ for event $i$ and $k = 0..m$. The choice of order ($m$=2 or $m$=3) is made heuristically for each site, with the aim of avoiding a reversal in gradient at sea-levels close to present-day (Fig. 2).

Not all events can be confidently identified in each $\delta^{18}O_b$ record, for example if a peak is not sufficiently resolved. However, provided multiple glacial and interglacial lowstands and highstands are included, our method is not very sensitive to the presence or absence of individual markers. We also note that our approach in compiling Table 1 was to obtain sufficient markers capturing a range of sites and methods, rather than carrying out an exhaustive review of all published sea-level estimates relevant to the last 800 kyr.

Once the transfer function has been used to convert $\delta^{18}O_b$ to $S$ (Fig. 3, middle column), $S$ is scaled to calculate changes in $\delta^{18}O_{ice}$:

$$\delta^{18}O_{ice} = S \times \Delta\delta^{18}O_{swLGM}/\Delta S_{LGM} \tag{4}$$

where $\Delta\delta^{18}O_{swLGM}$ and $\Delta S_{swLGM}$ are the global mean changes in $\delta^{18}O_{sw}$ and $S$ between the LGM and Holocene, taken by Bates et al. (2014) to be 1.0±0.1‰ and -130±10 m, respectively. Estimates of $\Delta\delta^{18}O_{swLGM}$ specific to the Southern Ocean suggest a slightly higher $\Delta\delta^{18}O_{swLGM}$=1.1±0.1‰ in that region (Adkins and Schrag, 2001; Adkins et al., 2002; Schrag et al., 2002; Malone et al., 2004). We continue to use the Bates et al. (2014) $\Delta S_{swLGM}$ = -130 ± 10 m which is consistent with several other recent studies (within errors) (Medina-Elizalde, 2013; Grant et al., 2014; Shakun et al., 2015; Hughes and Gibbard, 2018) (Table 1). Eq. 1 is then used to calculate the residual temperature contribution, i.e., $\delta^{18}O_T = \delta^{18}O_{ice} - \delta^{18}O_b$.

The third step converts $\delta^{18}O_T$ to water temperature $T_{sw}$, using a suitable paleotemperature equation. Bates et al. (2014) used the Shackleton (1974) *Uvigerina spp.* paleotemperature equation after applying suitable offsets for other species:

$$T_{sw} = 16.9 - 4.38(\delta^{18}O_T) + 0.1(\delta^{18}O_T)^2 \tag{5}$$

Five of the six sites selected in our study use *Cibicidoides* $\delta^{18}O_b$, and bottom water temperatures are frequently close to 0°C. Hence, we here use a more recent paleotemperature for *Cibicidoides* recommended by Marchitto et al. (2014, their Eq. 9), which covers temperatures down to -0.6°C:

$$T_{sw} = 111.36 - \sqrt{9392.77 + 909.09\delta^{18}O_T} \tag{6}$$

We also follow Marchitto et al. (2014) for *Uvigerina* $\delta^{18}O_b$, by again using the *Cibicidoides* Eq. 6 but after applying their recommended $\delta^{18}O_b$ offset of 0.47‰. This final step yields the bottom water temperatures shown in Fig. 3, which are then re-sampled to 4 kyr resolution as described in Section 3.4.

Each of the steps in this method requires assumptions that evidently introduce important uncertainties, as already discussed in detail by Siddall et al. (2010) and Bates et al. (2014). Perhaps most crucially, Bates et al. (2014) note that *"The calibration*

*windows for sea level are chosen as prolonged interstadial or stadial events, when sea level and temperature are at approx-imate equilibrium"* and that the method is not suitable during glacial inceptions and terminations. Some recent Pleistocene interglacials were likely too short, especially MIS 9 and 5, for all ice sheets to reach equilibrium with climate: this could require several thousands of years of constant climate (for example the Antarctic Ice Sheet: see Garbe et al., 2020). In contrast, deep ocean temperature responds to surface climate change at centennial time scales (Yang and Zhu, 2011; Li et al., 2013). Hence, the ambient temperature signal ($\delta^{18}O_T$) in $\delta^{18}O_b$ might respond to global climatic changes over time scales of order 0.1 kyr, while the ice volume $\delta^{18}O_{ice}$ signal likely responds over time scales reaching 10 kyr, potentially biasing reconstructions with this method during the rapid climate changes encountered through interglacials. However, if we assume that only the Greenland and Antarctic ice sheets remained during the warmer interglacials since MIS 11, the difference in ice volume between their transient and equilibrium states is likely less than ~12 m (based on complete loss of Greenland and West Antarctic ice sheets). That is only ~9% of the ~ 140 m glacial-interglacial amplitude of sea-level change, suggesting the assumption of "approximate" equilibrium is reasonable during interglacial calibration windows. Glacial calibration windows are selected near the ends of stadial conditions, when climatic changes are generally much slower or of lower magnitude than climatic changes during interglacials.

Further questions concern whether the scaling (Eq. 4) is regionally variable and whether it is linear. Regional variability may need considering for global studies; for example, $\Delta\delta^{18}O_{swLGM}$ is typically lower – $\sim 0.8\pm0.1‰$ – in the deep North Atlantic (Adkins and Schrag, 2001; Adkins et al., 2002; Schrag et al., 2002). However, four out of five estimates of $\Delta\delta^{18}O_{swLGM}$ at sites distributed around the Southern Ocean at 40 to 50 °S lie within $1.1\pm0.1‰$ (the fifth is $1.4\pm0.1‰$) (Adkins et al., 2002; Schrag et al., 2002; Malone et al., 2004), indicating little regional variability at least within that zone. The question of linearity arises because the oxygen isotopic composition of ice is not spatially uniform in ice sheets, so that $\delta^{18}O_{ice}$ does not necessarily vary linearly with $S$. This potential nonlinearity was considered by Bates et al. (2014) to introduce an error "of order 10%". Coarse-scale modelling by de Boer et al. (2012) supports a reasonably steady ratio of $\delta^{18}O_{ice}/S$ close to -1‰ per 100 m at 400 kyr time scales during the Pleistocene (see their Fig. 4), but with shorter time-scale variability reaching amplitudes of 0.3‰ per 100 m. Their results suggest the question of linearity should be revisited in future developments of this BWT method and when more detailed modelling results are available.

Overall, Bates et al. (2014) considered this method most appropriate for glacial-cycle time scales after the mid-Pleistocene transition (MPT), and less so for rapid changes during glacial inceptions or terminations, or prior to the MPT.

## 3.2 Additional uncertainties common to both methods

The relatively small variations in BWT over glacial-interglacial cycles have been recognised as a potential challenge in reconstructions (Tisserand et al., 2013; Stirpe et al., 2021). From an optimistic perspective we could see this lower variability as advantageous, since the less an environmental variable changes with time, the less concerned we need to be with reconstructing its changes. However, when considering the high sensitivity of ice shelf basal melt to small temperature changes (e.g. Burgard et al., 2022), we do indeed need accurate estimates of past CDW temperatures - even if these changes have been $< 1°C$.

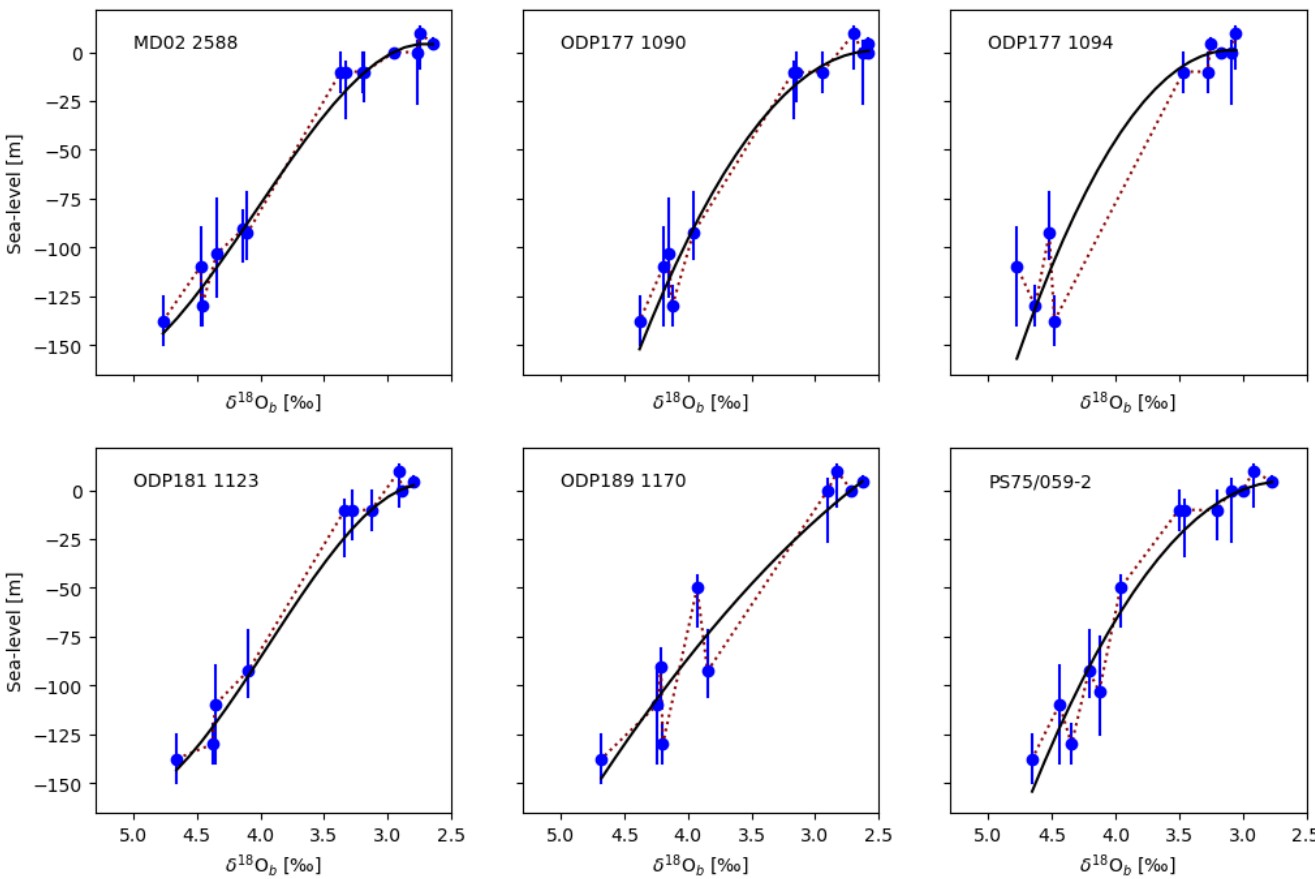

**Figure 2.** Transfer functions to convert $\delta^{18}O_b$ to sea-level (Section 3.1.2). Sea-levels and their uncertainty ranges (blue error bars) taken from the markers in Table 1 are plotted against $\delta^{18}O_b$ averaged over respective calibration windows in the $\delta^{18}O_b$ time series. Previously Bates et al. (2014) fitted linear piece-wise functions to a smaller selection of markers, which if applied here would yield the red dotted lines. Here we instead use second- or third-order polynomial fits (black solid line). Details of the sites and their selection are provided in Table 2 and Section 3.3.

Compared to planktic organisms used in SST reconstructions, benthic organisms are not subject to lateral advection while sinking, or to strong seasonality, or to varying habitat depth along a temperature gradient within the mixed layer and thermocline (Chandler and Langebroek, 2021a, and references therein). Nevertheless, some of the same problems do still apply – most notably related to calcite dissolution, sediment reworking, and dating uncertainties. These error sources are summarised briefly here, but more detailed discussion relevant to the specific sites and proxies used in this synthesis can be found in the original publications (Hodell et al., 2003a; Nürnberg et al., 2004; Elderfield et al., 2012; Bates et al., 2014; Ullermann et al., 2016; Hasenfratz et al., 2019; Starr et al., 2021; Stirpe et al., 2021).

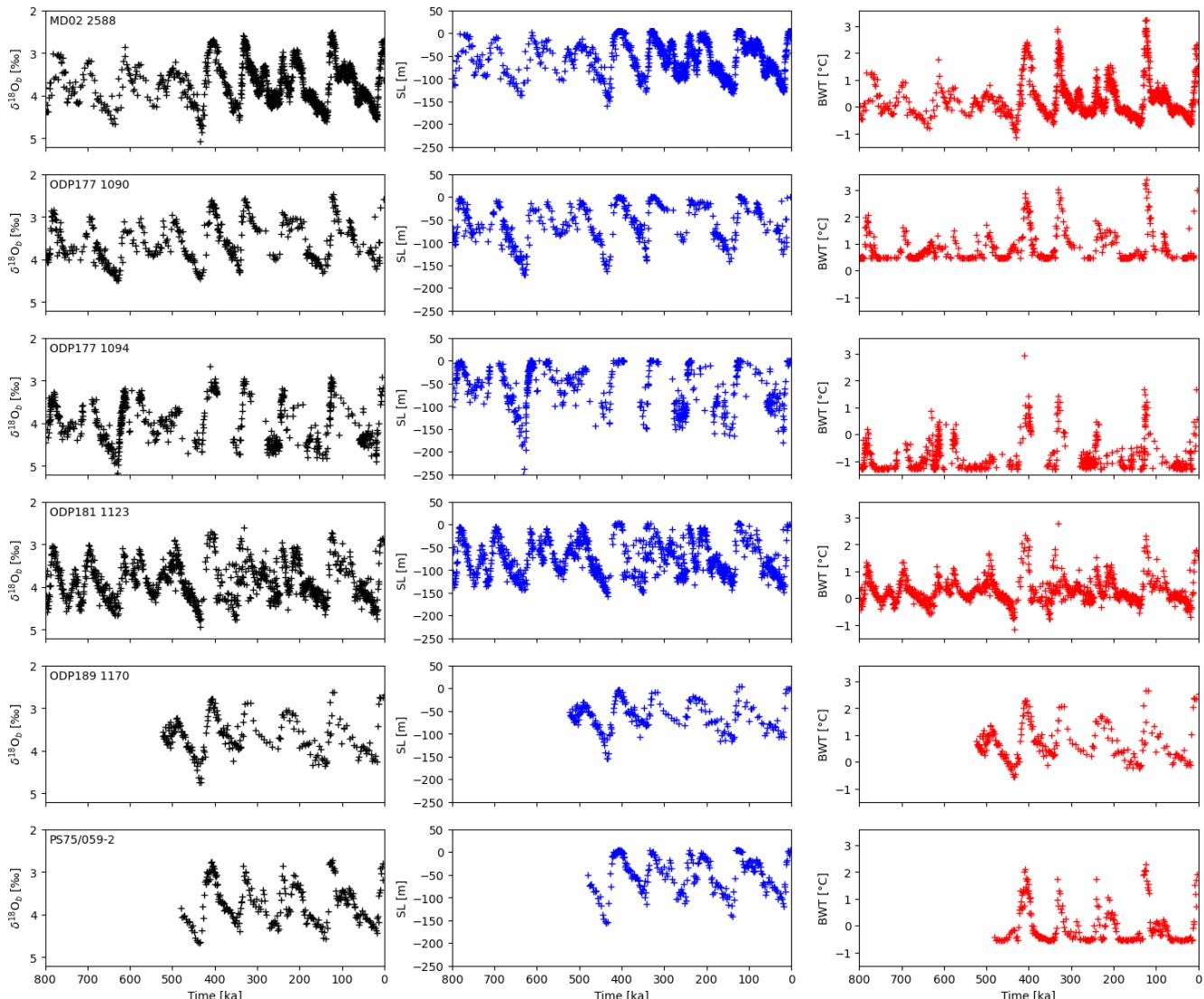

**Figure 3.** Extraction of the temperature signal from the oxygen isotopic content of benthic foraminiferal calcite ($\delta^{18}O_b$: see Section 3.1.2). Raw $\delta^{18}O_b$ time series (left column) are converted to sea-level time series (middle column) using site-specific transfer functions (Fig. 2). Sea-levels are then used to remove the ice volume contribution from $\delta^{18}O_b$, before applying a paleotemperature equation (Eq. 6) to calculate bottom water temperature (right column). Details of the sites and their selection are provided in Table 2 and Section 3.3.

Dissolution is an important problem for calcite-based proxies at deep ocean sites. Besides the loss of material available for proxy analysis, there is also less calcite available for dating (if using $\delta^{18}O_b$), potentially leading to lower resolution records. Dissolution rate generally increases with ocean depth, becoming most problematic below the foraminiferal lysocline (Berger, 1968). This lies at $\sim$ 3500 to 4000 m in the Southern Ocean (Williams et al., 1985; Hayward et al., 2001; Bostock et al., 2011),

only just reaching our deepest sites (Section 3.3), and is likely more significant for reconstructions of underlying AABW temperature than for CDW temperature. However, the lysocline may have been shallower during glacial periods, as carbonate
solubility increases at colder temperatures (Howard and Prell, 1994).

Further dating uncertainties arising from aligning one proxy record with another (e.g., benthic $\delta^{18}O_b$ with a global stack) can reach several kyr once outside the range of $^{14}$C dating – particularly for the lower resolution records. Dating uncertainties can also arise from subjective choice of tie-points in noisy records, and uncertainties in the target chronology. In a synthesis of records it must be noted that dating errors will tend to smooth out sharp peaks, if peaks that were in reality synchronous
across several sites are slightly asynchronous in the synthesis. On the other hand, by aligning our (mostly) benthic $\delta^{18}$O-based temperature reconstructions with the LR04 benthic $\delta^{18}$O stack, we may be imparting some artificial synchronicity – it is not clear to what extent these two factors will offset each other.

Post-depositional bioturbation causes vertical mixing of foraminifera shells in surface sediments, leading to observed age differences as high as 1 to 3 kyr between different foraminifera species in the same sediment horizon (Anderson, 2001; Broecker
et al., 2006; Mekik, 2014; Ausín et al., 2019). This vertical mixing is equivalent to temporal smoothing of the reconstructed temperature signal, and most strongly affects sites with lower sedimentation rates. Although bioturbation could be problematic if revising this synthesis to a higher temporal resolution in future when more data are available, we would not expect this error source to substantially alter our results at 4 kyr resolution. Reworking for example by winnowing and re-deposition adds a further source of error to both the reconstructed temperature and the age model, as local sediments become contaminated with
older material from distal sites (e.g., Dezileau et al., 2000).

As more records become available, site-specific assessments of the likely error sources would be worthwhile as an extra quality control step. Given the sparse data available at present we do not discriminate on this basis, or impose any weighting. Instead we assume the associated errors mainly contribute higher variance, rather than a substantial temperature bias.

### 3.3  Sites selection

CDW is an elusive water mass for paleotemperature reconstructions. As noted above, CDW lies above AABW, preventing the use of bottom water temperature (BWT) reconstructions from benthic organisms except where the bathymetry penetrates upwards through the AABW and into the CDW (for example, on the Chatham Rise: Elderfield et al., 2012). Meanwhile, CDW (and particularly lower CDW, of greatest consequence for Antarctic ice shelf melt: Wåhlin et al., 2010; Assmann et al., 2019) lies at depths beyond the reach of proxies based on surface or even sub-surface planktic organisms. Finally, poor preservation
affects some deep ocean sites (Section 3.2), while sediments in shallower water on the Antarctic continental shelf may have been repeatedly overridden by grounded ice during glacial stages, disturbed by deep iceberg keels, or are difficult to core due to logistical problems such as sea-ice. The above difficulties are perhaps compounded by a greater community interest in other environmental indicators (e.g., $\delta^{18}$O, $\delta^{13}$C, and planktic SST proxies). Even though suitable BWT proxies are available for the continental shelf (Hillenbrand et al., 2017; Totten et al., 2017; Mawbey et al., 2020), there are not yet any reconstructions
representative of CDW directly accessing ice-shelf cavities prior to the Holocene.

To-date, at glacial-interglacial timescales there only three temperature reconstructions representing CDW in the Southern Ocean (ODP sites 1123, 1090, 1094: Elderfield et al., 2012; Bates et al., 2014; Hasenfratz et al., 2019, respectively) (Fig. 4).

The only Antarctic Ice Sheet modelling studies using deep water temperature reconstructions directly are the GRISLI simulations by Quiquet et al. (2018) and Crotti et al. (2022), who used NADW temperatures reconstructed at ODP site 980 (55°N in the North Atlantic). Their use of this single distant site, and the current paucity of records from the Southern Ocean, raises the question of what geographic extent is appropriate if we wish to achieve reasonable statistical confidence and temporal resolution in reconstructed CDW temperatures. Even at coarse (multi-millennial) time scales, we need to find additional sites.

One option is to look further afield to the main water masses from which CDW is sourced, and assume these same water masses and regions have contributed in similar proportions to CDW under past climates. On that assumption, *changes* in these source-region water temperatures should be representative, to some extent, of *changes* in CDW temperature. This greater scope would expand our region of interest to include NADW in three key regions: the North Atlantic (including ODP 980 used by Quiquet et al. 2018 and Crotti et al. 2022); its transit south in the deep western boundary current which forms the lower, southbound leg of the AMOC (Lumpkin and Speer, 2007; Pardo et al., 2012; Rhein et al., 2015; Buckley and Marshall, 2016); and along its eastwards-spreading pathways near the equator and towards the Cape Basin in the South Atlantic (Smethie et al., 2000; Garzoli et al., 2015; Rhein et al., 2015). While this could yield several additional sites, there are important drawbacks to this approach. First, besides NADW, other deep water masses from the Indian and Pacific contribute to CDW, albeit more strongly to UCDW at present (Section 2). Second, temperature changes particularly in North Atlantic NADW are likely modified by subsequent mixing with overlying water masses or underlying AABW during transit southwards (Fig. 1). Third, a weakened AMOC during glacials could greatly reduce the contribution of NADW (or its glacial equivalent), making these sites less relevant during such periods. Overall, NADW temperature changes are interesting for comparison, but are not included in our synthesis.

Another option, which we follow here, is to analyse additional sites with $\delta^{18}O_b$ data that were not used by Bates et al. (2014). There are many such sites even in the Southern Ocean (e.g., see the database compiled by Mulitza et al., 2022). However, including all possible sites currently bathed by CDW would heavily weight the synthesis towards the last glacial cycle (particularly the last deglaciation), and would dominate any information provided by the two Mg/Ca records. Given that only long time-scale changes in BWT are captured by the $\delta^{18}O_b$ method (Section 3.1.2), and to preserve the diversity of proxies – albeit already limited to just two – we are selective in which additional records to include: these must lie south of 40°S; they must represent LCDW at present day; they must have sufficient resolution to establish sea-level calibration windows; and they must extend back to at least MIS 11, so that our results are not greatly affected by changing geographic coverage with time. This somewhat stringent selection yields five additional $\delta^{18}O_b$ BWT records (MD05-2588, ODP 1094, ODP 1123, ODP 1170, PS75/059-2; Table 2), bringing the total to six sites with eight temperature records (two Mg/Ca and six $\delta^{18}O_b$: Fig. 4 and Table 2). All the $\delta^{18}O_b$ records were analyzed for BWT following the method outlined in Section 3.1.2.

There is no guarantee that selected sites have been bathed in lower CDW throughout the last 800 kyr. This concern is discussed later in Section 5.3.

 **3.4 Bottom water temperature synthesis**

In total we use six sites located as described above (Table 2, Fig. 4). BWT at these sites has been reconstructed using benthic foraminiferal $\delta^{18}O_b$ and Mg/Ca. The records are published at varying temporal resolution, using different calibrations and chronologies. Therefore, to improve consistency and to considerably reduce variance, we stacked the records following a similar methodology to that employed in a SST synthesis covering the last 200 ka (Chandler and Langebroek, 2021b). Briefly there are five main steps:

1. Temperatures derived from *Uvigerina* Mg/Ca at ODP 1123 (Elderfield et al., 2012) were recalculated using the more recently published Stirpe et al. (2021) calibration (Mg/Ca $= 0.073T + 0.9$).

2. The reconstructed 'modern' temperature was subtracted so that down-core temperatures are anomalies from the recent past. Considering the low resolution of the records, the 'modern' temperature is an average of all samples younger than 2 ka (if possible) or otherwise the observed bottom water temperature reported by the original authors.

3. Age models were transferred to a consistent chronology by aligning $\delta^{18}O_b$ with the Lisiecki and Raymo (2005) LR04 global benthic stack.

4. All records were resampled to 4 kyr resolution following the method used by Chandler and Langebroek (2021b). This choice of coarse resolution reflects a necessary compromise between data resolution and uncertainty, because the uncertainty increases as resolution increases (due to fewer contributing sites). Note that our method admits gaps: we do not interpolate low-resolution records onto a higher resolution time series.

5. Uncertainties for each time slice were calculated using the t-distribution, under the assumption that temperatures in each record are sampled independently from those of other records. At the sediment cores for which both proxies are available, the two corresponding records are still treated as independent temperature estimates. This is reasonable, as in Section 5 we show it is the methodological errors rather than geographic variability that is contributing most variance. However, if the two records from one site are not fully independent, our confidence intervals will be slightly underestimated.

## 4 Results

Each of the 4-kyr time slices from 800 ka to present has BWT contributions from at least 3 sites, but mostly 6 or 7 (Fig. 5a-b). The mean BWT, which we consider to represent CDW temperature, shows clear glacial-interglacial variability, as well as the shift to stronger cycles after the Mid-Bruhnes event. Interglacial warming peaked in MIS 5 (0.5±0.5°C) and MIS 11 (0.6±0.4°C); MIS 9 was not well resolved at 4 kyr resolution. CDW temperatures only a little cooler than present-day were likely also reached in MIS 19 (-0.4±0.2°C) and 7 (-0.3±0.5°C). Confidence in warming during MIS 17 and 15 is low, due to higher scatter across fewer sites. Glacial cooling was similar through glacial stages 12, 10, 8, 6 and 2, with anomalies typically close to -2°C, and perhaps less severe (closer to -1.5°C) in glacials 18, 16 and 14.

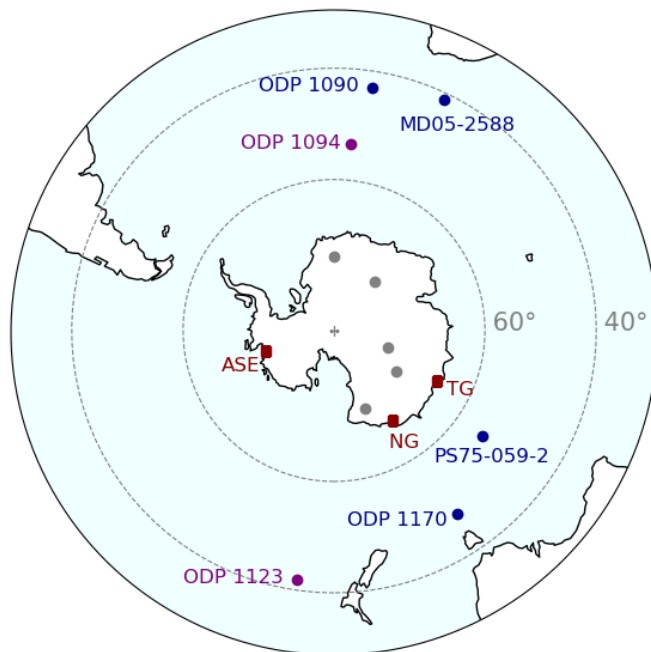

**Figure 4.** Locations of marine sediment cores used in our CDW temperature synthesis, and locations of other sites of note. Sediment cores with both Mg/Ca and $\delta^{18}O_b$ are coloured purple; the remainder with only $\delta^{18}O_b$ are coloured blue. Additional details of each site are provided in Table 2. Antarctic ice cores used in the Parrenin et al. (2013) surface air temperature reconstruction (grey circles) are, from top to bottom, EPICA Dronning Maud Land (EDML), Dome Fiji, Vostok, EPICA Dome C (EDC) and Talos Dome (TALDICE). Regions used in the discussion of ice shelf basal melt (red squares) are the Amundsen Sea Embayment (including Thwaites Glacier, ASE), Aurora basin (including Ninnis Glacier, NG) and Wilkes basin (including Totten Glacier, TG).

We observe some systematic biases between temperatures reconstructed with $\delta^{18}O_b$ and with Mg/Ca at the two sites for which both proxies are available (ODP sites 1094 and 1123: Fig. 7). The envelopes of points are very consistent between the two sites, in both cases showing that (i) the range of BWTs is wider for Mg/Ca than for $\delta^{18}O_b$, as also evident from the scatter in Fig. 5b; and (ii) Mg/Ca temperatures tend to be cooler than $\delta^{18}O_b$ temperatures in glacials, and warmer in interglacials, with a transition close to -1°C.

Comparing our synthesis with other Southern Hemisphere paleotemperature reconstructions, we find that CDW temperature closely follows both Southern Ocean SST and Antarctic surface air temperature (Fig. 5d-e) at 4 kyr time scales, but the amplitudes of glacial-interglacial $T_{CDW}$ changes are weaker. In both cases, the correlation is best described by a quadratic relationship, as the gradients ($dT_{CDW}/d(SST)$ and $dT_{CDW}/dT_{AIS}$) decrease at colder temperatures (Fig. 8). Close to present-day conditions ($SST = 0$ or $T_{AIS} = 0$), the respective gradients are 0.28 and 0.70, highlighting the much stronger changes in

Antarctic surface air temperature, and slightly stronger changes in Southern Ocean SST, than in $T_{CDW}$ during interglacials.

We also compare our $T_{CDW}$ reconstruction with NADW temperature, using $T_{NADW}$ calculated from $\delta^{18}O_b$ at site ODP 980 (Oppo et al., 1998; McManus et al., 1999; Flower et al., 2000). We chose ODP 980 as BWT at this site has been analysed previously (Waelbroeck et al., 2002; Bates et al., 2014) and used as an ocean temperature boundary condition in Antarctic Ice Sheet modelling (Quiquet et al., 2018; Crotti et al., 2022). Here we recalculate BWT following the same method as used above for $T_{CDW}$, except we use a slightly smaller $\Delta\delta^{18}O_{sw} = 0.8‰$ for the North Atlantic (instead of $1.1‰$ for the Southern Ocean), following Adkins and Schrag (2001) and Schrag et al. (2002). When compared with $T_{CDW}$ we find interglacial warming is similar or slightly weaker in $T_{NADW}$, while glacial cooling is consistently stronger in $T_{NADW}$ by 1 to 2°C (Fig. 5c). Overall the correlation is again described by a quadratic relationship (Fig. 8).

At global scale our $T_{CDW}$ is well correlated with a global mean deep water temperature (GDWT) reconstruction by Rohling et al. (2021) (Figs. 5e and 8). In this case the relationship is only weakly nonlinear (linear and quadratic $R^2$ values are very similar, at 0.66 and 0.67, respectively), and the magnitudes of interglacial warming appear closely matched between both records through most of the study period in Fig. 5. However, the glacial terminations and the peak interglacial warming both appear to occur earlier in GDWT than in CDW, and correlation is stronger ($r^2 = 0.83$) when a 4 kyr lag is applied to $T_{GDW}$ (Fig 9d).

Although further analysis of leads/lags in the above correlations is hindered by the coarse temporal resolution, it is neverthe-less worthwhile. Fig. 9 shows how the strength of correlation changes between $T_{CDW}$ and all four comparative paleotempera-ture records when leads or lags of up to 8 kyr are applied to $T_{CDW}$. For both $T_{AIS}$ and Southern Ocean SST, the lag of $T_{CDW}$ is likely close to 2 kyr, as lags of 0 and 4 kyr yield similar correlations (Fig. 9a,b). As noted above, the lag behind GDWT appears longer (close to 4 kyr: Fig. 9c). In contrast, $T_{CDW}$ appears to lead $T_{NADW}$ by 0 to 4 kyr.

# 5 Discussion

Considering the vast study area, and the two different proxies employed, there is not surprisingly considerable variance in BWT anomalies reconstructed at each time slice. The combination of high variance and a low number of sites, particularly in the period before ca. 500 ka, yields wide uncertainties for many of the time slices (Fig. 5). This variance is likely dominated by methodological errors, including non-thermal influences on Mg/Ca paleothermometry (e.g., Raitzsch et al. 2008, Stirpe et al. 2021, and Section 3.1.1), and uncertainties in sea-level estimates, transfer functions and additional assumptions required in the oxygen isotopic reconstructions (Siddall et al. 2010, Bates et al. 2014, and Section 3.1.2). To make this assessment we have first compared $T_{CDW}$ analysed with both proxies in the same core (i.e. where geographic variability is eliminated); here, root mean square differences are 0.7°C (ODP 1094) and 0.9°C (ODP 1123) (Fig. 7). We next compared $T_{CDW}$ regional averages in the Atlantic and Indian-Pacific sectors, which each conveniently have 3 $\delta^{18}O_b$ and 1 Mg/Ca records, thus minimising contributions of methodological errors; here the RMSD between groups is lower (0.5°C, Fig. 6). This comparison indicates methodological errors contribute considerably more variance than regional differences. Nevertheless, all RMSDs are substantially lower than glacial-interglacial variability, and the confidence intervals in the latter part of the study period (since 500 ka) are reasonably narrow. Therefore it is worthwhile in the next section to highlight close relationships with other paleotemperature records.

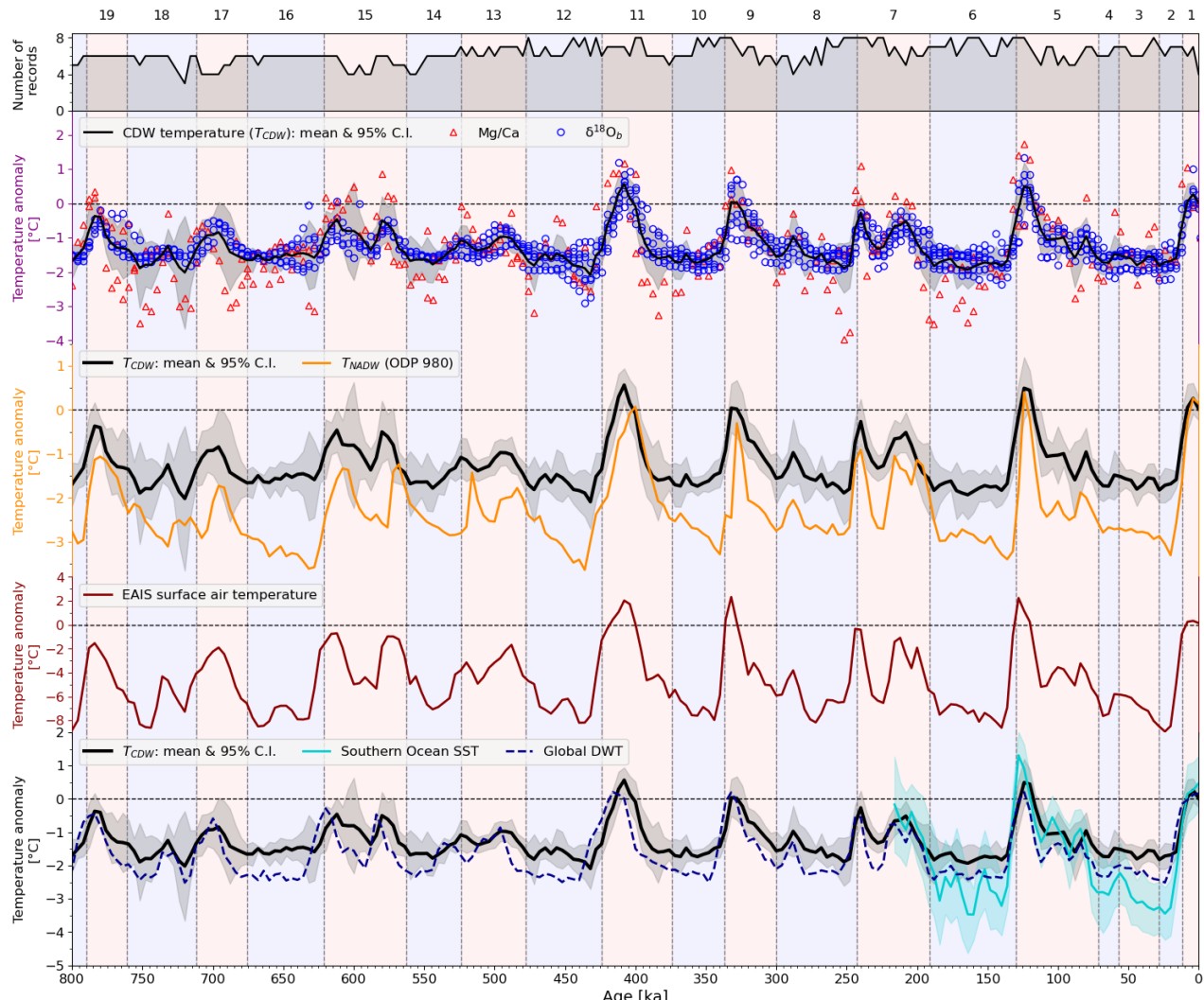

**Figure 5.** Time series of CDW temperature anomalies at the six sites in Table 2, compared with several other relevant records. (a,b) CDW temperature anomalies $T_{CDW}$ based on our bottom water temperature synthesis. Following the method in Section 3.4, records have been resampled to 4 kyr resolution, as shown by individual markers. The thick black line shows the mean of all $N$ records contributing to each time slice, with the 95% confidence interval (shaded grey) calculated using the t distribution (mean$\pm t\sigma/\sqrt{N}$). (c) Comparison between $T_{CDW}$ and NADW temperature anomalies ($T_{NADW}$). Here $T_{NADW}$ is estimated as the mean of two North Atlantic temperature reconstructions from $\delta^{18}O_b$ (ODP 980 and DSDP 607). (d) East Antarctic Ice Sheet surface air temp (SAT) from Jouzel et al. (2007) and Parrenin et al. (2013) (red), resampled to 4 kyr resolution to match the resolution of our BWT synthesis. (e) Southern Ocean SST from Chandler and Langebroek (2021b) (turquoise with shaded 95% confidence interval); and global mean deep water temperature from Rohling et al. (2021) (blue dashes). Both records are again resampled to 4 kyr. Numbers and vertical shading show the marine isotope stages (Lisiecki and Raymo, 2005).

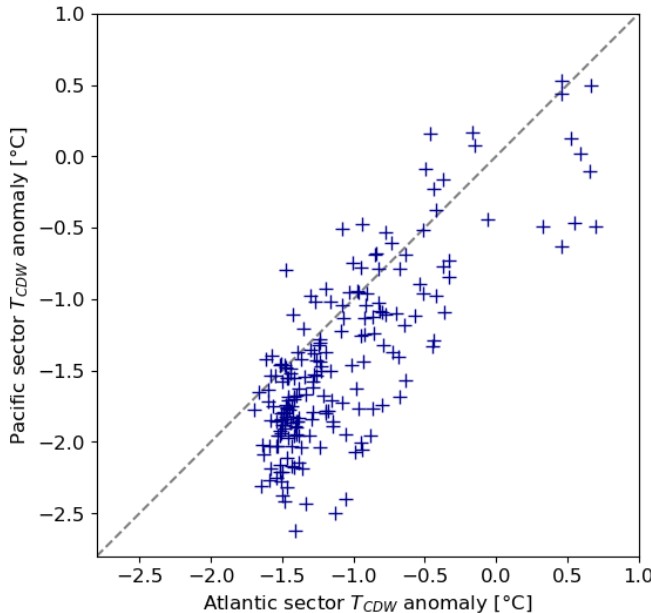

**Figure 6.** Correlation between mean $T_{CDW}$ in the three Atlantic sector cores and mean $T_{CDW}$ in the three Indian-Pacific sector cores (see Fig. 4 for locations). The dashed line indicates the 1:1 ratio. Each point represents a 4 kyr time slice.

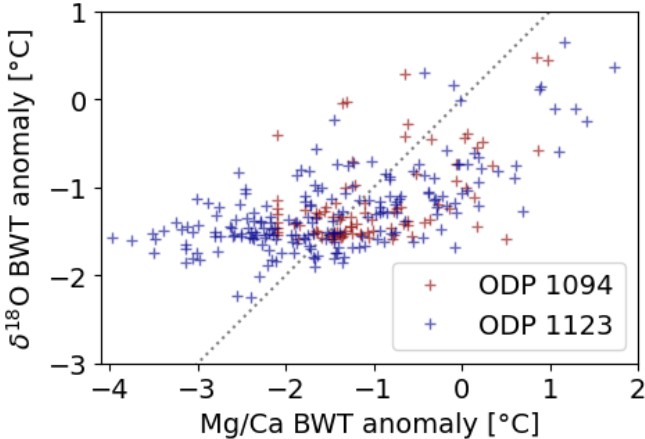

**Figure 7.** Correlation between BWT reconstructed with Mg/Ca and BWT reconstructed with foraminiferal $\delta^{18}O$, at the two Southern Ocean sites for which both are available. The dashed line indicates the 1:1 ratio. Each point represents a 4 kyr time slice.

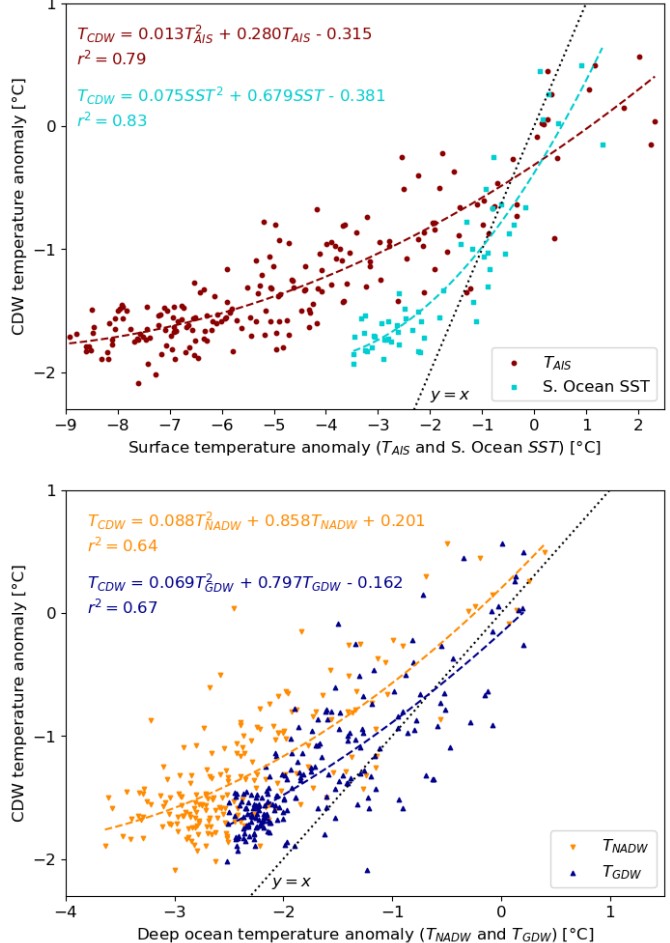

**Figure 8.** Correlation between our CDW temperature synthesis and other paleotemperature records. These are: (a) East Antarctic Ice Sheet surface air temperature ($T_{AIS}$; red) (Jouzel et al., 2007; Parrenin et al., 2013) and Southern Ocean sea surface temperature from 220 ka to present (SST; turquoise) (Chandler and Langebroek, 2021b); and (b) NADW temperature at ODP 980 ($T_{NADW}$, orange, this study) reconstructed using data from Oppo et al. (1998), McManus et al. (1999) and Flower et al. (2000), and global mean deep water temperature ($T_{GDW}$, blue) (Rohling et al., 2021). All records have been resampled to 4 kyr. Dashed lines and corresponding text show best fit curves calculated using quadratic regression. The black dotted line is $y = x$.

## 5.1 Regional and global context

Strong relationships emerge between our $T_{CDW}$ reconstruction and other paleotemperature records for the Southern Hemisphere and even globally (Figs. 5, 8). Indeed, the correlations of $T_{CDW}$ with East Antarctic Ice Sheet surface air temperature, Southern Ocean SST, ODP 980 NADW temperature, and global mean deep water temperature ($r^2$ values 0.79, 0.83, 0.64 and 0.67, respectively) are stronger than correlation between the two BWT proxies ($r^2 = 0.36$) (Figs. 7, 8). Consistent with

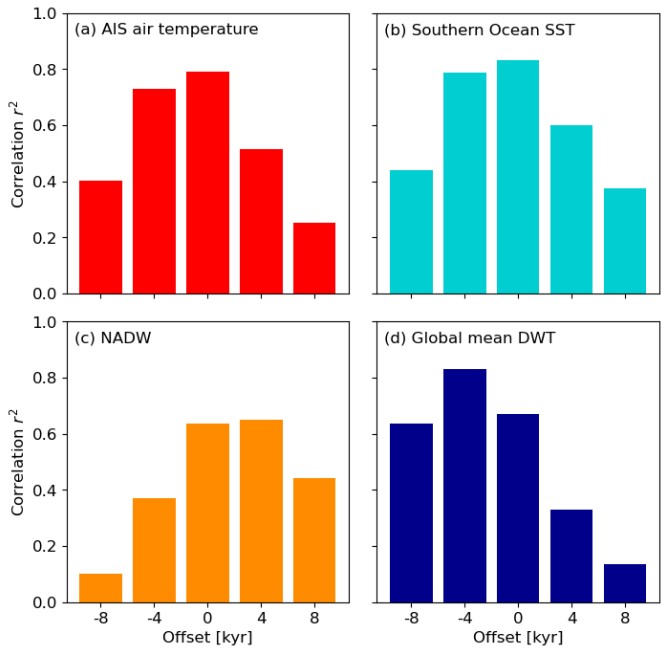

**Figure 9.** Correlation between $T_{CDW}$ and other paleotemperature records, after applying a temporal offset of -8 to +8 kyr to $T_{CDW}$. A negative offset (lag) indicates that changes in the other record *precede* changes in $T_{CDW}$. The four panels are for (a) East Antarctic Ice Sheet surface air temperature (Jouzel et al., 2007; Parrenin et al., 2013); (b) Southern Ocean sea surface temperature (for the period 220 ka to present, Chandler and Langebroek, 2021b); (c) NADW deep water temperature at ODP 980 (Oppo et al., 1998; McManus et al., 1999; Flower et al., 2000) (re-analyed in this study); and (d) global mean deep water temperature (Rohling et al., 2021). Since the $T_{CDW}$ time series is at 4 kyr resolution, leads/lags shorter than 4 kyr cannot yet be evaluated.

these close relationships are the magnitudes of interglacial warming, which are highest in MIS 11, 9 and 5 in our $T_{CDW}$
reconstruction as well as in $T_{AIS}$, $T_{NADW}$ and $T_{GDW}$.

    Although correlations are strong, magnitudes of glacial-interglacial changes in $T_{CDW}$ are typically weaker than in these other records, particularly when compared to surface temperature changes (Southern Ocean SST and East Antarctic Ice Sheet surface air temperature: Fig. 8a). This is due to both stronger interglacial warming and stronger glacial cooling in the surface temperature records, as indicated by the regression curves crossing the dotted line ($y = x$) in Fig. 8a. We would expect that
cooling of CDW (or of its equivalent glacial-state water mass) is limited by the physical lower bound imposed by the freezing point of seawater; this likely contributes to the decrease in gradients $dT_{CDW}/d(SST)$ and $dT_{CDW}/dT_{AIS}$ under glacial conditions (Fig. 8a). It is also likely that $T_{CDW}$ lags $T_{AIS}$ and SST – although probably by $< 2$ kyr, since correlations at 4 kyr applied lag are a little weaker than those with zero applied lag (Fig. 9a,b). This lag should be expected, given the lack of significant heat sources or sinks in the deep ocean, as temperature changes at depth must be driven primarily by changes in
surface heating and/or surface wind-driven circulation. Either way, a delay is introduced as these surface changes are propagated to the deep ocean by advection or diffusive processes. This is particularly relevant for CDW, which has no surface source region

and instead is a mixture of deep water masses. Under modern ocean circulation conditions the propagation of surface changes to the deep Southern Ocean is modelled to have time scales of order hundreds of years (Li et al., 2013; Yang and Zhu, 2011), consistent with the probably short ($< 2$ kyr) lag in our $T_{CDW}$ compilation. Asymmetry in the time scales for cooling and warming (e.g. Gough and Lin, 1992; Yang and Zhu, 2011; Li et al., 2022) is an interesting question that unfortunately cannot yet be evaluated at 4 kyr resolution.

The strong link between $T_{CDW}$ and surface temperature changes supports the use of surface temperatures (which are generally better constrained by proxy data) as a basis for parameterising deep ocean temperature *at this coarse 4 kyr temporal resolution*. The EDC ice core temperature reconstruction has been employed by paleo ice sheet modellers for example in parameterisations of sub-ice-shelf melting (Albrecht et al., 2020) or as a glacial index to interpolate linearly between ESM snapshots (Mas e Braga et al., 2021). In the former case, Albrecht et al. (2020) used ESM equilibrium simulations to quantify linear scalings between surface air temperature and deep ocean temperature, finding equilibrium $T_{CDW}$ was 0.75 times mean ocean temperature and 0.42 times $T_{AIS}$. The scalings compare similarly with our $dT_{CDW}/d(T_{GDW}) = 0.80$ at $T_{GDW} = 0°C$ but are somewhat higher than our $dT_{CDW}/d(T_{AIS}) = 0.28$ at $T_{AIS} = 0°C$. Hence, our quadratic relationship between $T_{CDW}$ and these other temperature indicators could be used to revise or validate scalings in these parameterisations. Implications of our results for ice shelf melt are discussed further in Section 5.4 below.

Through the last two glacial cycles (MIS 6 to present), our $T_{CDW}$ closely follows other recent global mean and deep water temperature estimates: this includes relatively steady temperature anomalies of ca. -2°C throughout the penultimate glacial (MIS 6) in our study, in global DWT (Shakun et al., 2015; Rohling et al., 2021) and in mean ocean temperature (Shackleton et al., 2020). This was followed by global deep and mean ocean temperature warming to ca. +1°C during the LIG (Shakun et al., 2015; Shackleton et al., 2020; Rohling et al., 2021). LIG warming is weaker (+0.5±0.5°C) in our BWT synthesis at 4 kyr resolution, but can be resolved to +0.8±0.7°C at 2 kyr resolution (albeit with less confidence: see Section 5.2 below). Consistent trends are then observed from MIS 5 to present in $T_{CDW}$ and $T_{GDW}$ (Shakun et al., 2015; Rohling et al., 2021). Interestingly the $T_{NADW}$ at ODP 980 is often cooler, particularly through glacial periods, and only reaches warming magnitudes comparable to $T_{CDW}$ in interglacials 9 and 5.

Such strong relationships between temperature records are encouraging as they indicate persistence of key underlying interactions and drivers of the climate system throughout the study period (800 ka to present), supporting the notion that paleoenvironmental reconstructions from any one or several glacial cycles within this period can provide valuable insights into our present-day interglacial climate. This confidence could extend to potential future changes, but only at multi-millennial timescales owing to the coarse temporal resolution we have used. However, given increasing evidence for close coupling between the Antarctic Ice Sheet and NADW in the North Atlantic (e.g. Iberian margin) even at millennial time scales (Hodell et al., 2023), we anticipate that strong relationships will emerge between $T_{CDW}$ and the records discussed above once $T_{CDW}$ can be reconstructed at higher resolution.

## 5.2 Temporal resolution and smoothing

One of the aspects we consider most problematic is the low resolution (4 kyr) of the synthesis. This is constrained partly, but not entirely, by the number of sites and the need to balance temporal resolution with statistical confidence. In all interglacials except MIS 15, there would still be sufficient records to resolve peaks with at least 6 sites if increasing resolution from 4 kyr to 2 kyr: in this case, some peaks become stronger because temporal smoothing is reduced. For example, the MIS 5 peak increases from 0.5±0.5°C (4 kyr) to 0.8±0.7°C (2 kyr), while MIS 11 increases only from 0.6±0.4°C (4 kyr) to 0.7±0.6°C (2 kyr). Further reducing the resolution to 1 kyr leads to fewer contributing records in each time slice, and very wide error bars. Even though 2 kyr resolution remains statistically viable through much of the study period, we chose not to use this higher resolution even for the latter period (MIS 11 to present) because it is not justified by assumptions underlying the $\delta^{18}O_b$ method, as outlined in Section 3.1.2. A higher resolution synthesis would need a greater number of Mg/Ca records, which do not have the same limitations on time scale.

Further smoothing is introduced in a regional mean temperature anomaly if spatial variability is not synchronous. This is increasingly less important at coarser resolution, because it only applies if the peak in one record is captured by a different time slice to the peak in another. However, with the sparse spatial distribution of records in our study, this is another argument against using a higher resolution, until there are sufficient records available such that the dataset can be subdivided into smaller regions.

For now we caution that 4 kyr resolution likely underestimates the magnitudes of peak warming in interglacials.

## 5.3 Persistence of CDW at selected sites

In Section 2 we provided a brief overview of the distribution of CDW and its source regions, that we used as a basis for site selection. However, there is no guarantee these selected sites will represent the same water masses in the past. Under climates that are increasingly different from present, the question of changes in ocean circulation becomes increasingly important. Of particular relevance is the extent to which the AMOC has weakened during glacial climates (e.g. Duplessy et al., 1988; Raymo et al., 1990; Yu et al., 1996; Marchitto et al., 2002; Lund et al., 2011; Gu et al., 2017; Kageyama et al., 2021; Kim et al., 2021). With a weaker AMOC, shoaling of the southbound flow of NADW would be accompanied by upwards expansion of AABW. This could have resulted in some deep sites in the Atlantic sector of the Southern Ocean being bathed in AABW, instead of upwelling NADW/CDW, during glacial climate states (Fig. 1). It is also possible that NADW was replaced to some extent by a different glacial water mass (glacial North Atlantic intermediate water: GNAIW Boyle and Keigwin, 1987; Duplessy et al., 1988; Marchitto et al., 2002). Glacial-interglacial circulation changes in the Pacific sector of the Southern Ocean were likely less variable, at least since the MPT (McCave et al., 2008; Bates et al., 2014).

Insights into water mass changes at our sediment core sites can firstly be gained by comparing the Atlantic and Indian-Pacific regional averages (Fig. 6). Here we find strong linear correlation ($R^2$=0.63), and the RMSD between regions is 0.5°C (∼20% of the glacial-interglacial variability). If a weakened glacial AMOC had strongly influenced the Atlantic sector sites, we might have expected to reconstruct greater cooling in any Atlantic sites switching to AABW. On the contrary, we find slightly weaker

glacial cooling in the Atlantic sector, on average, than in the Indian-Pacific sector. This is shown by points clustering below the dashed $y = x$ line in Fig. 6. On the other hand, $\delta^{13}$C at ODP 1090 (Hodell et al., 2003b) and MD02-2588 (Ziegler et al., 2013) suggest an increased influence of southern-sourced water masses during glacial periods. Our reconstructed glacial cooling in both regions (Atlantic and Indian-Pacific), is then interpreted as reflecting cooling of CDW in all sectors – likely driven partly by an increased contribution of AABW – rather than a switch to north-bound AABW at a subset of sites.

Further discussion of circulation changes is not warranted here but could be supplemented in future by other proxies analysed in the sediment cores we have used, e.g. $^{13}$C.

## 5.4 Implications for ice shelf basal melting

We next provide an indication of how our reconstructed temperature changes could *potentially* influence ice shelf basal melting, bearing in mind the important caveat that we have reconstructed changes in CDW temperature, and not changes in polewards CDW transport rate or modification by mixing across the continental shelf. Changes in either of these processes could further influence the temperature of CDW reaching ice shelf grounding lines. We use the Amundsen Sea Embayment in West Antarctica, and the ice shelves of the Wilkes and Aurora subglacial basins of East Antarctica as examples. Each of these regions has extensive basins grounded below sea-level and has been considered as susceptible to rapid, ocean-driven retreat under future warming (Arthern and Williams, 2017; Golledge et al., 2021; Reese et al., 2022; Jordan et al., 2023). Modified CDW currently accesses cavities beneath ice shelves in the Amundsen Sea and Aurora sectors (Jacobs et al., 1996; Wåhlin et al., 2010; Rintoul et al., 2016; Silvano et al., 2017; Wåhlin et al., 2021), but probably not in the Wilkes sector (Silvano et al., 2016).

With our anomaly-based temperature reconstruction ($T_{CDW}$), temporal changes in the thermal forcing ($T_F$) driving ice shelf basal melt are estimated using

$$T_F(x,y,t) = T_0(x,y) + \Delta T_{CDW}(t) - T_{PMP}(x,y) \tag{7}$$

where $T_0$ is the present-day temperature of water accessing sub-shelf cavities; $\Delta T_{CDW}$ is the CDW temperature *anomaly* we have calculated in this study, and $T_{PMP}$ is the pressure melting point at the ice shelf base, taken here as -1.8°C. When $T_F > 0$, ice shelf basal melt rates increase as ocean temperature increases (e.g. Beckmann and Goosse, 2003). When $T_F \leq 0$ there is no basal melting. Present-day temperatures ($T_0$) at 500 to 800 m depth adjacent to ice shelves in each region are extracted from the World Ocean Atlas (Locarnini et al., 2018) following Chandler et al. (2023), and closely match previous estimates by Schmidtko et al. (2014). Although transferring CDW temperature anomalies directly to changes in thermal forcing beneath ice shelves is a very simplistic representation of how ocean warming might translate to sub-shelf melting, this approach is commonly used by paleo ice sheet modellers due to an absence of practical alternatives (Quiquet et al., 2018; Albrecht et al., 2020; Sutter et al., 2019; Crotti et al., 2022), and at least illustrates regional differences in potential ice shelf susceptibility to interglacial warming.

Comparing the three regions (Fig. 10) we find three distinct characteristics in their $T_F$ time series. Thermal forcing in the Amundsen Sea Embayment remains positive throughout the 800 kyr study period, although the uncertainty envelope includes $T_F < 0$ in some glacial time slices. This indicates potential for persistent basal melting of ice shelves buttressing key ice streams

(e.g. Thwaites and Pine Island glaciers) even under glacial climates, provided that CDW can still access the continental shelf during glacial climate states. This would be consistent with a reconstructed LGM ice extent that did not reach the continental shelf break in the Amundsen Sea (Klages et al., 2017). Ice shelves buttressing the Aurora subglacial basin (e.g., Totten) are more likely to have switched from cold to warm conditions between glacials and interglacials, as $T_F$ switches from significantly negative to significantly positive. Finally the Wilkes subglacial basin is the only one of these three basins to have its main ice

shelves considered as a 'cold' cavities under present-day conditions (Silvano et al., 2016), and it thus seems likely to have remained in a cold state through much of the past 800 ka – except for the peaks of interglacials 11, 9 and 5.

Given the high sensitivity of the marine subglacial basins of West and East Antarctica to ocean-driven melting, the results from this very basic analysis highlight how we really need more CDW temperature reconstructions, to (i) increase confidence in glacial-interglacial temperature changes; (ii) improve temporal resolution, which is currently too poor for direct use of $T_{CDW}$

as a boundary condition; and (iii) evaluate regional variability. The issue of low resolution is likely to always hinder use directly as a boundary condition, because even decadal-scale temperature variability influences basal melt and grounding line migration (e.g. Jenkins et al., 2018). Hence in practice it is likely that stochastic noise, with characteristics determined based on modern observations and/or modelling, will need adding to a proxy-based temperature reconstruction.

For now, the synthesis would be more appropriately used to complement or validate the alternative (less direct) estimates

of CDW temperature changes employed in modelling studies (Pollard and DeConto, 2012; de Boer et al., 2014; Quiquet et al., 2018; Tigchelaar et al., 2018; Sutter et al., 2019; Albrecht et al., 2020), since none of these approaches has yet been independently validated.

## 5.5  Recommendations for comparison with climate model output

Besides its use as a boundary condition in ice sheet modelling, our synthesis can be used for comparison with climate model

output. This can be done in two ways: the first is to carry out site-by-site comparison; the second is to compute regional or water-mass averages in the climate model output and compare those with the regional average in this synthesis. The first method has the advantages that no water mass classification or regional averaging is required when processing the climate model output, and the site-by-site comparison reveals where the model-data match is particularly good or poor. A disadvantage is that we do not provide error bounds for individual sites. This is an important consideration since errors are likely to be large

for an individual site at each time slice, and only become reasonable when averaging over several sites. Errors particularly in the $\delta^{18}O_b$ method are hard to quantify and are instead combined with spatial variance in our empirical uncertainty estimates reported for the regional average. In the case that site-specific errors were needed, we would recommend conservative estimates of errors in each step, that can be propagated through the whole analysis in a Monte-Carlo simulation.

## 5.6  Priorities for future reconstructions

Given the high value that CDW temperature reconstructions have for ice sheet and climate modellers, whether for model validation or as a boundary condition, further efforts to improve the resolution, reduce variance and add regional variability would be very welcome. These could be addressed by:

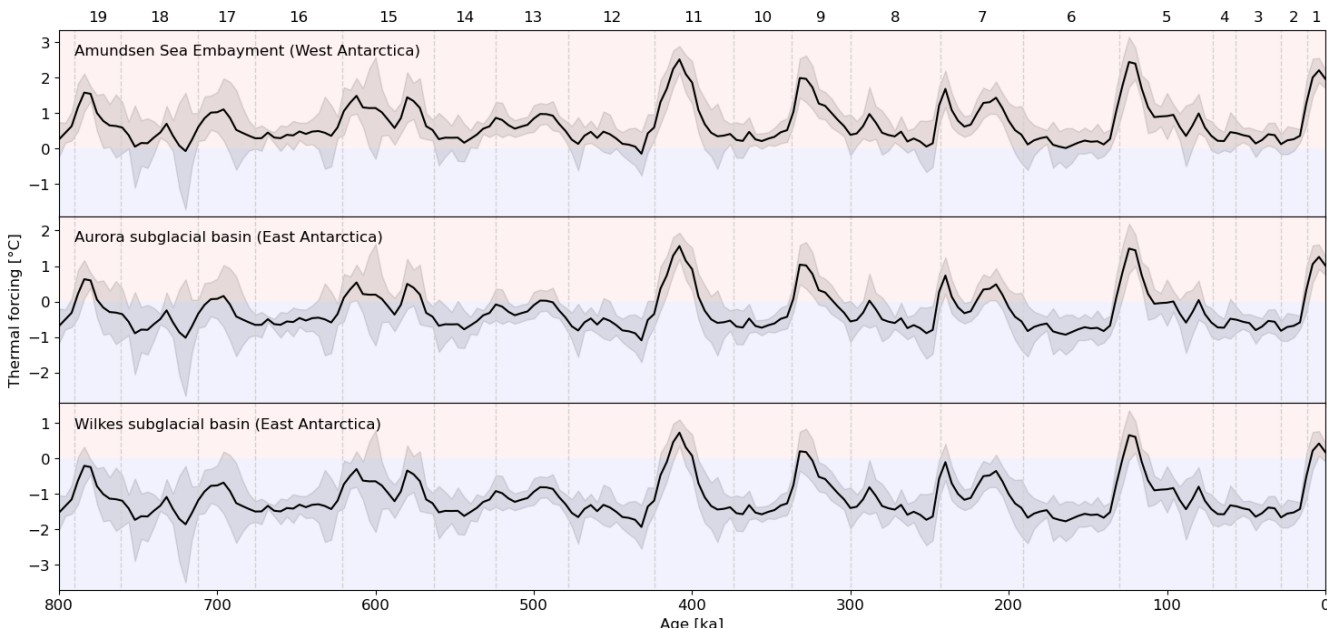

**Figure 10.** Estimated changes in thermal forcing for ice shelf basal melt in the Amundsen Sea, Aurora and Wilkes sectors of the Antarctic Ice Sheet. Here we consider thermal forcing as the temperature of upwelling CDW ($T_{CDW}$), before it is modified during its transport across the continental shelf and into ice shelf cavities. In this very simplistic demonstration there is no guarantee that CDW always accesses sub-shelf cavities or that transport rates have not changed, i.e., an increase in thermal forcing only indicates *potential* for increasing ice shelf basal melt under ocean circulation conditions conducive for CDW access to cavities.

1. Analysis of foraminiferal Mg/Ca in additional Southern Ocean sediment cores. Even relatively low-resolution records, which might have limited use individually, can be helpful in a synthesis of reconstructions provided reasonable age control can be established. This is because we use the t-distribution to calculate confidence intervals, and the width of confidence intervals decreases rapidly with increasing sample size, when sample size is small.

2. Further work to account for the influence on non-thermal factors in Mg/Ca paleothermometry, in particular carbonate chemistry, and how these might be corrected for in the Southern Ocean (Bryan and Marchitto, 2008; Healey et al., 2008; Raitzsch et al., 2008).

3. Application of additional proxies, in particular those less sensitive to carbonate chemistry: for example clumped isotope paleothermometry (e.g. Tripati et al., 2010; Piasecki et al., 2019; Peral et al., 2022) or Li/Mg (Bryan and Marchitto, 2008; Chen et al., 2023). Calcium isotope paleothermometry has also shown potential in benthic foraminifera, but less convincingly at temperatures below 3°C (Gussone and Filipsson, 2010; Mondal et al., 2023). An increased diversity in proxy types helps reduce bias associated with a single method or organism, and can provide empirical validation (for example, if one proxy shows very different trends from the others).

4. Further application of multiple proxies to the same core samples would be greatly beneficial, to quantify proxy biases and uncertainties independently of variance contributed by dating uncertainties and geographic variability. Only two sites in this synthesis have both proxies, and at these sites it appeared the methodological errors contributed more variance than spatial variability.

## 6 Conclusions

Here we have synthesised BWT reconstructions in the Southern Ocean with the aim of estimating changes in CDW temperature from 800 ka to present. Although BWT reconstructions are sparse in comparison to SST, there are sufficient data to establish a statistically meaningful synthesis at 4 kyr resolution. This yields CDW temperature anomalies of ca. -2 to -1.5°C during glacial periods, warming to 0.6±0.4°C and 0.5±0.5°C during the strongest interglacials (MIS 11 and 5, respectively) (Fig. 5). The MIS 7 CDW temperature anomaly was comparatively cooler at -0.3±0.5°C, and MIS 9 was poorly resolved at our temporal resolution.

There are many periods of high uncertainty, particularly prior to MIS 11, attributed to the combination of high variance amongst a small number of sites. The high variance is likely dominated by methodological rather than geographic variability (see Section 4 and Figs. 6 and 7). Despite the uncertainty, we find strong correlation with the AIS surface air temperature and Southern Ocean SST at time-scales of 4 kyr and longer, with evidence for a likely short (< 2 kyr) lag of $T_{CDW}$ behind $T_{AIS}$ and SST (Figs. 5, 8, 9). We also find very close agreement between our CDW temperature estimates and the Rohling et al. (2021) global DWT reconstruction, but with the latter leading by up to 4 kyr. Interestingly there is a stronger relationship with GDWT than with $T_{NADW}$ at ODP 980 (Fig. 8). Our results do not provide evidence of the strength of such correlations at shorter timescales, and additional sites that help to increase the temporal resolution would be very beneficial in this respect.

Given the importance of the deep ocean in both climate variability and Antarctic Ice Sheet mass balance, an increase in the number of Southern Ocean sites with CDW temperature reconstructions is urgently needed, for its own worth in understanding past oceanographic changes and also for use by modellers as a boundary condition or for validation.

*Data availability.* Original temperature reconstructions are available from the sources cited in Table 1. The synthesis is available at https://zenodo.org/doi/10.5281/zenodo.13253696 or on request from the authors.

*Author contributions.* DC compiled the synthesis. Both authors contributed to the manuscript.

*Competing interests.* The authors declare no competing interests

*Acknowledgements.* This study was funded by the European Union's Horizon 2020 research and innovation programme under grant agreement no. 820575 (TiPACCs). We gratefully acknowledge the authors of the original temperature reconstructions cited in Table 2, who have made their data publicly available. We also thank the Editor and two anonymous reviewers, whose comments have greatly improved this paper.

| Event | Sea-level (B14) [m] | Sea-level (this study) [m] | Sources |
|---|---|---|---|
| Mid-Pliocene | +28 [+12,+44] | – | S10 |
| MIS 12 | – | -138 [-150, -125] | G14, H18, S15 |
| MIS 11c | – | +10 [-8, +13] | G14 |
| MIS 10 | – | -103 [-125, -75] | G14, S15 |
| MIS 9e | – | 0 [-26, +6] | G14, M13 |
| MIS 8 | – | -93 [-106, -72] | G14, S15 |
| MIS 7e | -10 [-15, -5] | -10 [-20, 0] | G14, M13 |
| MIS 7c | -12.5 [-15, -10] | – | S10 |
| MIS 7a | -12.5 [-15, -10] | -10 [-25, 0] | G14 |
| MIS 6 | – | -110 [-140, -90] | D15, G14, S15 |
| MIS 5e | +8.7 [+8, +9.4] | +5 [+2, +7] | G14, D15, D20 |
| MIS 5c | -11.3 [-12, -10.6] | -10 [-20, 0] | G14, M13 |
| MIS 5a | -11.3 [-12, -10.6] | -10 [-34, -5] | G14, M13 |
| MIS 4 | -90 [-100, -80] | -91 [-107, -81] | S15 |
| MIS 3 | -70 [-90, -50] | -50 [-70, -44] | G14, M13, S15 |
| MIS 2 | -130 [-140, -120] | -130 [-140, -120] | G14, H18, M13, S15 |
| MIS 1 | 0 | 0 [-2, 0] | D15 |

**Table 1.** Sea-level markers used for establishing transfer functions. Markers used by Bates et al. (2014) are provided here for comparison. Sources and very briefly the methods employed are as follows: C17 Creveling et al. (2017) global database of high-stands in MIS 7a, 7c; D15 Dutton et al. (2015) review of global evidence; D20 Dyer et al. (2021) Bahamas paleo shore-lines and GIA modelling; G14 Grant et al. (2014) Red Sea benthic oxygen isotopes; H18 Hughes and Gibbard (2018) review of ice extent, marine oxygen isotopes and ice core evidence; M13 Medina-Elizalde (2013) global compilation of coral benchmarks; R06 Rabineau et al. (2006) sedimentary evidence (Western Mediterranean); S10 Siddall et al. (2010); S15 Shakun et al. (2015) paired benthic/planktic oxygen isotopes.

| Site | Location | Proxy | Proxy ref |
|------|----------|-------|-----------|
| MD02-2588 | 41.2S 8.9E, 2905 m | *C. wuellerstorfi* $\delta^{18}O_b$ | Starr et al. (2021) |
| ODP 1090 | 42.9S 8.9E, 3702 m | *C. wuellerstorfi* $\delta^{18}O_b$ | Hodell et al. (2003b) |
| ODP 1094 | 53.2S 5.1E, 2807 m | *M. pompiloides* Mg/Ca, *Cibicidoides spp.* $\delta^{18}O_b$ | Hasenfratz et al. (2019) |
| ODP 1123 | 41.7S 171.5W, 3290 m | *Uvigerina spp.* Mg/Ca (*), *Uvigerina spp.* $\delta^{18}O_b$ | Elderfield et al. (2012) |
| ODP 1170 | 47.2S 146.1E, 2704 m | *C. wuellerstorfi* $\delta^{18}O_b$ | Nürnberg et al. (2004) |
| PS75-059-2 | 54.2S 125.4E, 3613 m | *Cibicidoides spp.* $\delta^{18}O_b$ | Ullermann et al. (2016) |

**Table 2.** Proxy records used for bottom water temperature reconstructions. Notes: (*) BWT recalculated here using the Stirpe et al. (2021) calibration for *Uvigerina spp.* Mg/Ca = 0.073T + 0.9.

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
