# Peer review of "Glacial-interglacial Circumpolar Deep Water temperatures during the last 800,000 years: estimates from a synthesis of bottom water temperature reconstructions"

_EGUsphere, 2023_

## Author Comment (AC1)

**Summary**

We would like to thank the reviewers for their critical but constructive comments, which will be addressed in a revised manuscript.

**Reviewer #1**

**(1.1) The premise on the ice sheet melting is not really dealt with - I was expecting the authors to get back to it at the end and show how their new records might link to it.**

A similar comment was made by Reviewer 2 (see Point 2.5 below), so it would seem readers expected us to use our results to discuss implications for ice sheet response or at least ice shelf basal melting in more detail. This was not the original intention – our aim was to compile existing temperature records that are potentially suitable for estimating CDW temperature changes (Lines 62-68). The results would then be used by others, as a transient boundary condition for ice shelf basal melt parameterisations (for example with PICO, Reese et al., 2018), or for validating alternative estimates (e.g., a glacial index, Sutter et al., 2019; or linear response function, Albrecht et al., 2020). In the Introduction section of the revised paper we will make the objectives and scope clearer, while also adding further discussion in Section 5 as follows.

In Section 5.5 ('Recommendations for use as a boundary condition or for model validation") we already discuss some implications/limitations of our results from a modelling perspective, but this section can be expanded. We could add more discussion about the link to ice sheet modelling and particularly (i) the recent use of the N. Atlantic ODP 980 NADW temperatures as a boundary condition in some previous Antarctic Ice Sheet modelling (Quiquet et al., 2018; Crotti et al., 2022) and (ii) a persistent lack of independent validation of ocean temp estimates in past AIS modelling studies over glacial cycle time scales (de Boer et al., 2014; Tigchelaar et al., 2018; Albrecht et al., 2020; Sutter et al., 2022). This study could provide such validation – as we have done for our recent ice sheet modelling study with PISM (Chandler et al., in review) – at least at multi-millennial time scales.

As there are no ice sheet simulations in this study, we cannot use our results to estimate changes in shelf basal melting directly. However, calculation of changes in thermal forcing for some example ice shelves in Antarctica provides some interesting insights that can be discussed, as follows.

Here we consider the Wilkes and Amundsen Sea regions, which have both been considered susceptible to marine ice sheet instability in the future or in past interglacials. The thermal forcing plotted in Fig. 1.1 below, is:
$T_F = T_0 + \Delta T_{CDW} - T_{PMP}$,
where $T_0$ is the regional-average present-day ocean bottom water temperature at 500-800 m depth; $\Delta T_{CDW}$ is the CDW temperature anomaly we have calculated in this study, and $T_{PMP}$ is the pressure melting point, taken here as -1.8 degC. When $T_F$ is positive, ice shelf basal melt rates increase as temperature increases. When $T_F$ is negative there is no basal melting. Present-day temperatures for each region are extracted from the World Ocean Atlas (Locarnini et al., 2018) and closely match previous estimates by Schmidtko et al. (2014) for both regions. Although transferring CDW temperature anomalies directly to thermal forcing at grounding lines is very simplistic representation of how ocean warming might translate to sub-shelf melting (as alluded to in Lines 354-359), this approach has some precedent (Sutter et al., 2019; Albrecht et al., 2020) and illustrates important regional differences in likely ice shelf susceptibility to interglacial warming.

Even with our wide uncertainty envelope, we find $T_F$ in the Wilkes region is significantly negative for extended periods during glacials, and ambiguous ($T_F = 0$ lies within the error envelope) during most interglacials. In contrast, thermal forcing in the Amundsen Sea region is ambiguous through all the glacials and significantly positive in interglacials 19, 11, 9, 7, 5 and 1. Overall this highlights how we really need to reduce uncertainties in reconstructed CDW temperature – not only during interglacials, but also during glacials. Importance of the latter is illustrated by the Amundsen Region: here the LGM

is often used as a spin-up or initial state for ice sheet models, yet the ocean temperature uncertainty envelope encompasses both $T_F$ < 0 degC (negligible melt) and $T_F$ close to 1 degC (substantial melt). During model tuning and optimisation these two scenarios would likely lead to quite different modelling choices, in particular relating to parameterisations of calving and sub-shelf melting, which in turn would impact simulated responses to future warming. The same problem affects the penultimate glacial maximum (PGM; MIS 6), with uncertainties in sub-shelf melt and ice volume during the PGM in turn feeding into LIG sea-level estimates (Dendy et al., 2017).

We also note the coarse time scale (4 kyr) likely misses short-term warming events and will dampen interglacial peaks. As we discuss below in Point 1.2, this cannot easily be remedied by adding additional d18O sites – it would require either additional Mg/Ca records or application of a novel proxy (perhaps clumped isotopes).

[Figure]

**Figure 1.1.** Thermal forcing in two regions considered susceptible to marine ice sheet instability: Wilkes (Cook, Ninnis, Mertz ice shelves) and Amundsen (Pine Island, Thwaites, Dotson ice shelves). In both regions, ice shelf melt is currently driven by incursion of CDW onto the continental shelf.

**(1.2) Furthermore it is unclear to me why they focussed on the last 800 ka - they didn't really try to interpret anything older than 400 ka for the interglacials. Why only 7 records - there are others in the literature for LCDW already published that go back to ~ 800 ka e.g. ODP 1168, ODP 1170, ODP 1171 from the South Tasman Rise, south of Tasmania (Nurnberg et al., 2004). I would suggest the authors look at the new dataset of d18O recently published by Mulitza et al., 2022 ESSD, which would have many more records covering the last 400 and even 800 ka. Given the lack of records that go back 800 ka why not focus on the shorter time periods – especially when there are no periods warmer than present between MIS11 and MIS19. They compare to several other datasets that do not cover the last 800 ka in their discussion.**

Study period
This was chosen for several reasons, noted briefly at Lines 65-67 (we can make this clearer), as follows. (i) We focus on the period dominated by 100 kyr glacial cycles after the MPT, which is the climate state best characterising our present-day interglacial - albeit before the very significant anthropogenic influence. (ii) To match the longest Antarctic ice core record (EPICA Dome C). (iii) Importantly, we find added interest of including colder interglacials (rather than starting from MIS 11), since we can build a clearer picture of Earth system response to warming by including cooler as well as warmer interglacials. We will emphasize this more throughout the text.

Additional records
As a synthesis paper, our original scope was to compile records for which bottom water temperatures had already been published. There are of course far more benthic d18O records, than the few analysed by Bates et al. (2014) – as evidenced by the recent efforts of Mulitza et al. (2022) and previously by the common use of benthic d18O in establishing age models. We were originally

hesitant to include these d18O records without published bottom water temperatures for several reasons, but agree with the reviewer that it might be useful to perform a more full analysis.

Our main concern relates to the method to convert benthic foraminiferal d18O (d18Ob) to BWT, which relies on establishing site-specific transfer functions between sea-level and d18Ob (Siddall et al., 2010; Bates et al., 2014). The transfer functions are then used to separate the two main influences on d18Ob in the paleotemperature equation (sea water d18Osw, which is closely related to ice sheet ice volume and thus sea-level; and the ambient seawater temperature during the growth phase). The transfer functions are linear piece-wise functions established using "calibration windows" – typically full glacial or interglacial conditions – for which both sea-level and d18Ob data are available. Crucially, Bates et al. note that "*The calibration windows for sea level are chosen as prolonged interstadial or stadial events, when sea level and temperature are at approximate equilibrium*" and caution that the method is not suitable during glacial inceptions and terminations – as we have noted in Lines 307-312. Unfortunately, the interglacials are likely too short for ice sheets to reach approximate equilibrium with climate, as this would require tens of thousands of years of constant temperature and sea-level (for example the Antarctic Ice Sheet: see Garbe et al., 2020). In contrast, deep ocean temperature responds to surface climate change at centennial time scales (Yang et al., 2011; Li et al., 2013). Hence, the ambient temperature signal in d18Ob might respond to global climatic changes over time scales of order 0.1 kyr, while the ice volume (d18Osw) signal likely responds over time scales of 10 kyr, potentially biasing reconstructions with this method during the rapid climate changes encountered through interglacials. Adding extra sites based on d18O can help improve the signal to noise ratio at our current 4 kyr temporal resolution, and can help by targeting a more geographically relevant region, but we would essentially be reinforcing an underlying bias during interglacials (which are often the periods of most interest). This same limitation also prevents us from justifying an increased temporal resolution even if there are sufficient data to do so from a statistical aspect. Consequently, these extra Southern Ocean sites could certainly benefit a 800 kyr coarse resolution synthesis, but we would not gain much useful information by attempting a higher resolution synthesis over a shorter period e.g. since MIS 11.

Bearing in mind the potential pros and cons of analysing additional d18O records, we have carefully reviewed ca. 130 benthic d18O records south of 40 degS in the Mulitza database. For our purposes, suitable records need to cover at least one complete glacial cycle to enable a transfer function to be established. They also need to represent a suitable water mass (ideally lower CDW). If we also include a few additional ODP sites seemingly missed from that database we find ~28 suitable d18O records, of which only one (ODP 1090) was included in our original synthesis. To estimate temperatures at these sites we would have to establish transfer functions at each site, for which we could use more recent sea-level estimates than those used by Bates et al. (2014; their Appendix A). Evidently the selected sea-level estimates should be independent of global benthic d18O.

We would need to note that a synthesis with many d18O sites and only two Mg/Ca sites (assuming we exclude the North Atlantic sites), will be heavily biased towards the d18O method.

For the two Southern Ocean sites with Mg/Ca (and potentially the Atlantic sites M16672 and Chain 82-24-4PC), calculation of BWT using both Mg/Ca and d18O would provide an interesting comparison.

In a revised manuscript, we plan to first analyse the five suitable d18O sites that extend through the full study period from 800 ka to present (MD02-2588; ODP1090,1094,1123; PC493) as this subset represents all three main Southern Ocean basins and includes the two sites with long Mg/Ca records (ODP 1094,1123). The subset also includes the valuable Antarctic continental margin site PC493. Further details are provided in Table 1 below. We could also analyse an additional five sites with data from MIS 11 onwards (FR1/94-GC3, MD07-3076&3077; ODP1168, 1170; PS75-059-2). Discussion of results could then focus on (1) the full 800 ka period based on the seven long records; (2) extra interesting details, if any, that emerge in the shorter period (after 424 ka) with twelve records; (3)

comparison between the two proxies, which is limited as before by the small number of sites with Mg/Ca; and (4) comparison with selected North Atlantic records including ODP 980 used for AIS modelling by Quiquet et al. / Crotti et al., and the equatorial Mg/Ca record M16772 for NADW used in our original synthesis.

The limitations of the d18O method as described above will also be explained in more detail in the Methods section.

**Table 1:** Suggested list of sites to include. Sites in **bold** cover the full 800 ka study period, and sites in normal type at least the period since the start of MIS 11 (424 ka).

| Site | Lat/Lon/Depth [deg, m] | Max age [ka] | Basin | Proxy | Hydrography | Proxy ref |
|------|------------------------|--------------|-------|-------|-------------|-----------|
| **MD02-2588** | -41.2, 25.5, 2905 | 1656 | IND | d18O (Cbw) | NADW (Hall et al., 2018). | Starr et al. (2021) |
| **ODP 1090 & TTN057-6-PC4** | -42.9, 8.9, 3702 | 2903 | ATL | d18O (Cbw) | LCDW/NADW (Hodell et al., 2003). (Glacial GNADW? Howe et al., 2016). | Hodell et al. (2003) |
| **ODP 1094** | -53.2, 5.1, 2807 | 1557 | ATL | d18O (Cbs) Mg/Ca (Mp) | LCDW (Hasenfratz et al. 2019). (Glacial GNADW? Howe et al., 2016). | Hasenfratz et al. (2019) |
| **ODP 1123** | -41.8, -171.5, 3290 | 1546 | PAC | d18O (Us) Mg/Ca (Us) | LCDW (McCave et al., 2008). | Elderfield et al. (2012) |
| **PC493** | -71.1, -119.9, 2077 | 800 | PAC | d18O (Cbw) | LCDW (Williams et al., 2019) | Williams et al. (2019) |
| FR1/94-GC3 | -44.3, 150.0, 2667 | 454 | PAC | d18O (Cbw) | LCDW (Moy et al., 2008; Struve et al., 2022). | DeDeckker et al. (2018) |
| MD07-3076 & 3077 | -44.2, -14.2, 3777 | 440 | ATL | d18O (Cbs & Us) | LCDW (Gottschalk et al., 2016) (Glacial GAABW? Howe et al., 2016). | Gottschalk et al. (2016, 2019) |
| ODP 1168 | -42.6, 144.4, 2463 | 500 | IND | d18O (Cbw) | LCDW (Moy et al., 2008; Struve et al., 2022). | Nürnberg et al. (2004) |
| ODP 1170 | -47.2, 146.1, -2704 | 460 | IND | d18O (Cbw) | LCDW (Moy et al., 2008; Struve et al., 2022). | Nürnberg et al. (2004) |
| PS75-059-2 | -54.2, -125.4, 3613 | 480 | PAC | d18O (Cbs) | LCDW (Ullerman et al., 2008). | Ullerman et al. (2016). |

Ocean basins: ATL, IND, PAC Atlantic, Indian, Pacific. Foraminifera taxa: Cbs Cibicdoides spp.; Cbw Cibicidoides wuellerstorfi; Cs Cibicides spp.; Mp Melonis. pompiloides; Us Uvigerina spp.

**(1.3) Additionally, the new compilation shows strong agreement to past DWT compilations - so I'm not entirely sure that is really adds much to the literature. I was really hoping it would get into how it might influence the ice sheets or modelling so that it did add something new.**

It is true that our synthesis agrees closely with the Rohling et al. (2021) global mean DWT reconstruction, when their reconstruction has been resampled to match our 4 kyr resolution (see our original Fig 3). Note there was poorer agreement with the Shakun et al. (2015) global DWT (which showed consistently warmer interglacials). Good agreement with Rohling et al was not an inevitable outcome, given the different geographic focus, and we view this as an interesting result rather than a weakness. We would also note the agreement is strong at the coarse 4 kyr resolution that we used, but as more Mg/Ca data become available and a higher resolution synthesis is feasible, it is of course possible that periods with weaker agreement will emerge.

Regarding the influence on ice sheets or modelling, please see our response to Point 1.1 above.

**(1.4) I also find the inclusion of the NADW records (4 out of the total of 7) a little odd given the focus on the LCDW and evidence in the literature of the shut down of AMOC and therefore I would not expect the NADW to be a strong contributor to LCDW during the glacials. The authors came to this eventually in the paper, but only after putting it all together and didn't really discuss the implications on their records which have 4 records from NADW that may well swamp the LCDW datasets. Although I realise the authors are interested in the interglacials for the ice sheet melting. But then didn't really come back to this and how the MIS5, 9 and 11 may have impacted the ice sheets.**

It is indeed unintuitive to include NADW (particularly from N Atlantic sites) - this decision was originally motivated by the use of the North Atlantic ODP 980 BWT anomaly directly as an ocean temperature boundary condition in Antarctic Ice Sheet modelling (Quiquet et al. / Crotti et al.). We need to be clearer about this point in the Introduction and/or Section 2. In our original version we already discussed this issue of whether NADW records should be used, and compared the records for CDW and NADW, which it seems are well correlated – there is actually better agreement between the NADW and CDW subsets of records, than between the d18O and Mg/Ca subsets (original Fig 4). The scope of that comparison is rather limited by the low number of respective sites.

We are aware that sites currently bathed in NADW could have become bathed in a southern-sourced water mass (SSW) during glacial stages. However, northbound AABW (interglacials) or SSW (glacials) both eventually recirculate to mix with southbound NADW (interglacials) or GNAIW/GNADW (glacials). See Fig 1.4 below, with schematics copied from Howe et al. (2016) and Matsumoto (2017), as well as Ferrari et al. (2014). Assuming the Atlantic remains a relatively important heat source for CDW, temperature changes at sites intermittently bathed in deep northbound water masses remain relevant even during glacial periods, and should still reflect changes in CDW temperature albeit with more delay than for southbound water masses. The extra delay is not a critical issue at 4 kyr resolution. Nevertheless, it would be good to avoid reliance on such arguments.

Following Point 1.2 above, in the revised paper we aim to restrict our synthesis to records south of 40degS, by analysing (or re-analysing) the d18O data in some selected cores (Table 1). However, because (i) ODP 980 has recently been used by the ice sheet modelling community, (ii) data from the North Atlantic could help boost the sparse data in the Southern Ocean, and (iii) there is potential interest in the changing contribution of NADW to CDW, we prefer to keep some discussion and comparison of the North and South Atlantic NADW records even if they are excluded from the main synthesis.

[Figure]

**Figure 1.4.** Two schematics comparing modern and glacial ocean circulation patterns in the Atlantic. The left hand pair is copied from Howe et al. (2016) and is based mainly on $\varepsilon_{Nd}$. The right hand pair is copied from Matsumoto (2017), and is largely based on carbon isotopic analysis by Keigwin and Swift (2017). Throughout a glacial cycle, bottom water in the North Atlantic (south of approx. 60N, and below ~2000 m) eventually mixes with water masses upwelling in the Southern Ocean (CDW, or a glacial equivalent) contain contributions from southbound NADW/GNAIW in the upper overturning cell as well as AABW returning south in the lower cell. Therefore, temperature changes in NADW/GNAIW as well as AABW should all eventually influence the temperature of upwelling CDW.

**(1.5) Many sections there was insufficient detail on the methods and the issues – they came back to many of these in the discussion at the end – but they should have been upfront. There really was insufficient detail around the methods used for the Mg/Ca and the d18O. I realise these are in the original papers – but you need to provide a summary here so the reader doesn't have to go back to the original papers.**

We are happy to add more details on both proxies used (d18O and Mg/Ca) and will move information on their limitations from the Discussion to the Methods (Section 3). In particular we will include some issues related to the d18O proxy (Point 1.2 above) and carbonate dissolution (Point 1.8 below).

**(1.6) I felt the oceanography background needed more information and detail on the CDW – and differentiate between the NADW, LCDW, UCDW and mCDW – it would have helped to have a 3D/Depth figure to show the links between these water masses.**

Rather than a 3D figure could we suggest complementing Fig 1 with vertical sections of the Atlantic sector and SW Pacific, showing key circulation features, as used in many similar studies (similar to the examples copied above for Point 1.4). We can also refer to original definitions of these water masses.

**(1.7) It was unclear to me how the authors plotted up Figure 4 when you can't look at BWT on both the NADW and the LCDW at the same time? How do you make plots like this? Also which cores had both benthic d18O and the Mg/Ca done on them to make the other X-Y plot.**

Fig 4a in the manuscript compares the averages of BWTs reconstructed from all cores with Mg/Ca (Chain 82-24-4PC, M16772, ODP 1094, ODP 1123), to the averages of BWTs calculated from all cores with d18O (DSDP 607, ODP 980, ODP 1090). Each point represents one 4 kyr time slice. There were no cores with both proxies.
Similarly for Fig 4b there is one point for each time slice. NADW BWT anomaly is averaged over cores currently bathed in NADW (Chain 82-24-4PC, DSDP 607, M16772, ODP 980, ODP 1090). CDW BWT anomaly is averaged over cores currently bathed in CDW (ODP 1094, ODP 1123).
This will be clarified in the caption.

**(1.8) The reason why there are not many DWT records using Mg/Ca for benthic foraminifera is that it is not a trivial thing to do. Firstly there are not always sufficient tests of the right species of benthic foraminifera. Secondly the method is time consuming and has some issues – some of which were outlined in the limitations. But one thing that has not been mentioned in the paper is the potential impact of dissolution and the Carbonate Compensation Depth - the LCDW of the Southern sits at or very close to the Carbonate Saturation Horizon which is around 3000 m and the CCD which sits around 4000 m- so not many cores actually preserve sufficient Carbonate organisms or the records can be compromised by period of dissolution due to the shoaling of the CCD - during the glacial cycle.**
Yes the preservation issue was mentioned only very briefly (calcite dissolution) and not specifically linked to CCD, which is indeed an important limitation on where samples can be collected. This aspect will be discussed in more detail and we will refer to original studies to check for reports of carbonate dissolution at each site.
We will also move details of the limitations from the Discussion to the Methods.

Chandler and Langebroek compiled 7 deep water records (3 benthic Mg/Ca, 1 ostracod Mg/Ca, and 3 benthic d18O) during the last 800 ka, aiming to understand the glacial-interglacial temperature variability of Circumpolar Deep Water. However, there are several problems with data selection, data interpretation, and presentation, I have to say that this study, at least in the current form, is not to the standard of the CP.
Here are my main concerns.

**(2.1) The premise of using a compilation of BWT at selected sites to reflect CDW BWT change. Including NADW sites in the compilation is not appropriate because 1) NADW sites could be bathed in different water masses during glacials and interglacials due to circulation change, as already mentioned by the authors in Section 5.3, 2) shallow NADW sites (e.g. ODP 980) would have minimal influence on CDW that can potentially upwell underneath the Antarctic Ice shelves, due to the lower seawater density. On the contrary, Pacific sites at relatively deep depths, though downstream of CDW, may more reliably record the CDW temperature as those could be on the same isopycnal as CDW upwelling underneath the Antarctic Ice shelves. More benthic d18O records from the Pacific included in Bates et al (2014) thus might be included in the compilation as well. But ultimately, it is not most convincing to include sites not bathed in CDW to infer CDW changes. There may be more SO sites available if the time span of the compilation can be shortened.**

As we have noted in response to Reviewer #1 (Point 1.4 above), the use of NADW is certainly not intuitive and was originally motivated by the use of ODP 980 directly as an ocean temperature boundary condition in some recent Antarctic Ice Sheet simulations (Quiquet et al., 2018; Crotti et al., 2022). It does seem that deep water masses in both the N and S Atlantic have contributed to upwelling CDW in the Southern Ocean through glacial as well as interglacial climates, just less directly in the latter case (see Point 1.4 and copied schematics above). However, it would clearly be better to not include the N. Atlantic records.

In our revised synthesis we intend to analyse additional d18O records from the Southern Ocean, including some 'downstream' locations in the SW Pacific (Table 1 above). It was not our original intention to analyse additional records – but it will greatly help – please see our more detailed response to Point 1.3, as there are several restrictions limiting the number of suitable sites.

**(2.2) Using a compiled CDW T to inform ice shelf melting triggered by CDW.**

**The authors set out to use a compiled CDW T record to infer potential ice shelf melting triggered by CDW during previous warm interglacials. It is noted that such an event can be triggered by ~1 degC warming in a short period (a year) in the modern ocean (e.g., Jenkins et al., 2018), which is a really small difference challenging for any given record reconstructed by any proxy to resolve with confidence, let alone a compilation smoothing multiple records. So the 0.1-0.5degC warming during previous interglacials mentioned in the abstract must be interpreted in the context of the uncertainty range, which is not reported but can be expected to be large enough to make the reported warming statistically insignificant.**

Here the comment "*The authors set out to use a compiled CDW T record to infer potential ice shelf melting triggered by CDW during previous warm interglacials*" was beyond our original stated intention of compiling a temperature synthesis (Lines 63-64). We had not intended to use our synthesis to estimate changes in ice shelf melting. However, it seems that this was not clear, and we will add some related discussion (see Point 1.1 above).

It is true that the strong sensitivity of basal melt to temperature changes of 1 degC is a challenge for using proxy records – or indeed any other existing method – for estimating basal melting. We already admit the 4 kyr resolution record is too coarse and too uncertain, at present, to be used directly, and

that is why we recommend its use for validation of alternatives rather than directly as a boundary condition (Lines 347-353). The high uncertainty is not in itself a reason to not attempt this compilation, and one of our aims of this study is to highlight how the uncertainty is still high, in the hope this stimulates community interest in acquiring more data. Of course, we acknowledge that will not be an easy task! Owing to the problems highlighted above with converting benthic d18O to BWT we can suggest efforts should be directed towards additional Mg/Ca records, or to clumped isotopes which may be feasible in the Southern Ocean and at relevant temperatures (Peral et al., 2018; Leutert et al. 2021), but have not yet been applied at time scales relevant to this study. Regional averages over several sites remain the best way at present to overcome substantial uncertainties at individual sites.

Regarding *Warming in the interglacials is not statistically significant and should have been reported with uncertainty intervals in the abstract:* Since we intend to add more records, we hope that the error bounds will be reduced. Either way we will report error bounds more consistently.

Finally, the short time scales: it seems very unlikely that we will be able to reconstruct annual or decadal temperature changes, over such a long time period. This is a potential issue when for example transition from one state to another is influenced by stochastic noise as well as longer term changes in climate forcing (e.g., Niu et al. 2019). We suspect the best approach will be to use observations, such as those from Jenkins et al. (2018) and several others, to characterise short term variability as noise that can be superimposed on the millennial-scale ocean temperature changes. A similar approach is used for surface climate forcing in ice sheet models. We can add this point to the discussion.

**(2.3) Hydrographic settings of CDW.**

**As a manuscript on CDW, there are lots of inaccurate statements about CDW. While NADW is the heat source of the modified CDW, it is not the principal source water mass of CDW (Line 7). Deep waters from the Indian and Pacific sectors, as well as deep water formed around Antarctica, are also key sources of CDW (refer to Talley 2011 for detailed hydrography). Also, UCDW and LCDW are not defined properly. And modified CDW is often used in the context of melting under ice shelves, but it is not a counterpart of UCDW and LCDW.**

Thanks, we will correct these inaccuracies in the revised manuscript and provide some definitions.

**(2.4) Lack of details about their method.**

**how the BWT is calculated? How is the sampling done? Which types of uncertainties are included in the uncertainty envelope shown in Figure 4? These questions are not trivial. Details, such as choice of d18O-T equation, treatments of ice volume change, and local effects on seawater d18O, are lacking for benthic d18O calculation. For the sampling method, no original data is shown, so one would have no way to assess if the compilation is potentially biased by one particular site.**

Some additional information on the d18O method is provided in response Point 1.1 above, and follows Bates et al. (2014). The methods of calculating BWT from Mg/Ca and d18O will be provided in more detail in the Methods, and the respective limitations (currently in the Discussion) will also be moved to the Methods section. We will also add a figure with subplots showing temperature reconstructions at each individual site.

Not sure what is the error envelope alluded to in Fig 4 (scatter plots) – is the reviewer referring to Fig 2? In that case the error envelope for each time slice is calculated using the t-distribution, i.e., as mean +/- t*sigma/sqrtN, using all N samples for that time slice.

Individual sites are shown in Figure 2.4 below, using blue crosses for the original data and red circles for the 4 kyr resampled data. The modern water temperature is shown by the dashed black line. A tidier version of this figure can be included in the manuscript.

[Figure]

**Figure 2.4:** Sites used in the original manuscript, showing original data (blue crosses), 4 kyr resampling (red circles), and present-day water temperature (dashed line).

**(2.5) The structure of the paper.**

**A reader is left with the impression that the work done here did not at all contribute to solving the research question raised in the introduction, i.e., the potential influence of CDW T on ice shelf melting during previous interglacials. The introduction sets the expectation of a reader very high and by finishing reading I felt a little disappointed. Perhaps, the authors could put something that can be achieved by this work in the introduction. Also, I think it would be better to provide some background on the limitations of the employed proxies in the introduction.**

Regarding the aims and scope of the paper in terms of influence on ice shelf melting, we will add some discussion on thermal forcing - please see our response to Point 1.1 above.

Limitations were not dealt with in sufficient detail in our original manuscript (Section 5.4) and will be described in more detail – in particular with regard to the assumptions underlying the d18O proxy. We will provide these in the Methods section, as we will also be providing more information on the two proxies in that section.

References

Albrecht et al. (2020), Cryosphere, https://doi.org/10.5194/tc-14-599-2020.
Bates et al. (2014), QSR, https://doi.org/10.1016/j.quascirev.2014.01.020.
Chandler et al. (in review), preprint at https://www.researchsquare.com/article/rs-3042739.
Crotti et al. (2022), Nat. Comms, https://doi.org/10.1038/s41467-022-32847-3.
de Boer et al. (2013), Clim. Dyn., https://doi.org/10.1007/s00382-012-1562-2.
Dendy et al. (2017), QSR, https://doi.org/10.1016/j.quascirev.2017.06.013.
DeDeckker et al. (2018), Pangaea datasets https://doi.org/10.1594/PANGAEA.893207,
        https://doi.org/10.1594/PANGAEA.893210.
Ferrari et al. (2014), PNAS, https://doi.org/10.1073/pnas.1323922111.
Garbe et al. (2020), Nature, https://doi.org/10.1038/s41586-020-2727-5.
Gottschalk et al. (2016), P&P, https://doi.org/10.1002/2016PA003029 and Pangaea datasets
        https://doi.org/10.1594/PANGAEA.858257, https://doi.org/10.1594/PANGAEA.858257.
Gottschalk et al. (2019), Pangaea dataset https://doi.org/10.1594/PANGAEA.898191.
Hall et al. (2018), AGU Fall Meeting abstract.
Hasenfratz et al. (2019), Science, https://doi.org/10.1126/science.aat7067 and Pangaea dataset
        https://doi.org/10.1594/PANGAEA.901384.
Hodell et al. (2003), G3, https://doi.org/10.1029/2002GC000367 and Pangaea datasets
        https://doi.org/10.1594/PANGAEA.218129, https://doi.org/10.1594/PANGAEA.218131.
Howe et al. (2016), Nat. Comms., https://doi.org/10.1038/ncomms11765.
Jenkins et al. (2018). Nat. Geos. https://doi.org/10.1038/s41561-018-0207-4.
Keigwin and Swift (2017), PNAS, https://doi.org/10.1073/pnas.1614693114.
Leutert et al. (2021), Clim. Past, https://doi.org/10.5194/cp-17-2255-2021.
Li et al. (2013), Clim. Dyn., https://doi.org/10.1007/s00382-012-1350-z.
Locarnini et al. (2018), WOA18, https://www.ncei.noaa.gov/access/world-ocean-atlas-2018/.
Malone et al. (2004), EPSL, https://doi.org/10.1016/j.epsl.2004.02.027.
Matsumoto (2017), PNAS, https://doi.org/10.1073/pnas.1701563114.
McCave et al. (2008), QSR, https://doi.org/10.1016/j.quascirev.2008.07.010.
Moy et al. (2008), JQS, https://doi.org/10.1002/jqs.1067.
Niu et al. (2019), GRL, https://doi.org/10.1029/2019GL083717.
Nürnberg et al. (2004), Geophys. Mono. Ser., https://doi.org/10.1029/151GM17.
Peral et al. (2018), GCA, https://doi.org/10.1016/j.gca.2018.07.016.
Quiquet et al. (2018), GMD, https://doi.org/10.5194/gmd-11-5003-2018.
Schmidtko et al. (2014), Science, https://doi.org/10.1126/science.1256117.
Siddall et al. (2010), QSR, https://doi.org/10.1016/j.quascirev.2009.05.011.
Starr et al. (2021), Nature https://doi.org/10.1038/s41586-020-03094-7 and Pangaea dataset
        https://doi.org/10.1594/PANGAEA.921111.
Struve et al. (2022), Nat. Comms., https://doi.org/10.1038/s41467-022-31116-7.
Sutter et al. (2019), Cryosphere, https://doi.org/10.5194/tc-13-2023-2019.
Tigchelaar et al (2018), EPSL, https://doi.org/10.1016/j.epsl.2018.05.004.
Ullerman et al. (2016), P&P https://doi.org/10.1002/2016PA002932 and Pangaea dataset
        https://doi.org/10.1594/PANGAEA.833421.
Yang et al. (2011), GRL, https://doi.org/10.1029/2011GL048076.

---

## Author Response (AR1)

**Response to reviewers' comments**

**Summary**

We would like to thank the reviewers for their critical but constructive comments, which we have now addressed in substantial revisions to all sections of the manuscript. In this revised version we have abandoned the use of North Atlantic sites in our compilation, and instead analysed new sites in the Southern Ocean. This new analysis has considerably reduced the error bounds in our results. We do however still make a comparison between our compilation and North Atlantic site ODP 980. Specific comments are addressed below, **with the reviewers' comments in bold type** and our replies in normal type. Since most of the manuscript has been rewritten we refer the reviewers to specific sections of the manuscript relevant to each comment rather than copying extensive parts of the revised text into this responses document.

**Reviewer #1**

**(1.1) The premise on the ice sheet melting is not really dealt with - I was expecting the authors to get back to it at the end and show how their new records might link to it.**

A similar comment was made by Reviewer 2 (Point 2.5 below), so it would seem readers expected us to use our results to discuss ice sheet response, or at least basal melting, in more detail. This was not the original intention – our aim was to compile existing temperature records that are suitable for estimating CDW temperature changes (Lines 62-68). The results would then be used by others, as a transient boundary condition for ice shelf basal melt parameterisations (for example with PICO, Reese et al., 2018), or for validating alternative estimates (e.g., a glacial index, Sutter et al., 2019; or linear response function, Albrecht et al., 2020). In the Introduction section of the revised paper we have made the objectives and scope clearer [Lines 70-77]. Nevertheless, some discussion of implications for ice shelf melt is useful, and we have added this in a new Section 5.4 "Implications for ice shelf basal melting". This new analysis shows some interesting differences between thermal forcing estimates in three sectors of Antarctica (new Fig 10).

**(1.2) Furthermore it is unclear to me why they focussed on the last 800 ka - they didn't really try to interpret anything older than 400 ka for the interglacials. Why only 7 records - there are others in the literature for LCDW already published that go back to ~ 800 ka e.g. ODP 1168, ODP 1170, ODP 1171 from the South Tasman Rise, south of Tasmania (Nurnberg et al., 2004). I would suggest the authors look at the new dataset of d18O recently published by Mulitza et al., 2022 ESSD, which would have many more records covering the last 400 and even 800 ka. Given the lack of records that go back 800 ka why not focus on the shorter time periods – especially when there are no periods warmer than present between MIS11 and MIS19. They compare to several other datasets that do not cover the last 800 ka in their discussion.**

Study period
This was chosen for several reasons, noted only briefly at original Lines 65-67. We have now clarified this choice [Lines 64-69]: (i) it covers the period for which there are sufficient data to establish a meaningful synthesis at a practical resolution; (ii) it covers the 100 kyr glacial cycles most relevant to our present climate state, albeit before the onset of anthropogenic influences; (iii) inclusion of colder interglacials prior to MIS 11 can build a clearer picture of Earth system response to warming; (iv) it matches the duration of the longest ice core record (EPICA Dome C: Jouzel et al. 2007); and (v) proxy records derived from oxygen isotopes (the main data source here - see methods) are considered less reliable prior to the MPT (Bates et al. 2014).

Additional records

As a synthesis paper, our original scope was to compile records for which bottom water temperatures had already been published. There are of course far more benthic d18O records, than the few analysed by Bates et al. (2014) – as evidenced by the recent efforts of Mulitza et al. (2022) and previously by the widespread use of benthic d18O in establishing age models even in the Southern Ocean. The extent to which we should expand the scope to include the analysis of new sites, should consider the following points.

(i)   The method to convert benthic foraminiferal d18O (d18Ob) to BWT relies on establishing site-specific transfer functions between sea-level and d18Ob (Siddall et al., 2010; Bates et al., 2014). The transfer functions are then used to separate the two main influences on d18Ob in the paleotemperature equation (sea water d18Osw, which is closely related to ice sheet ice volume and thus sea-level; and the ambient seawater temperature during the growth phase). The transfer functions are linear piece-wise functions established using "calibration windows" – typically full glacial or interglacial conditions – for which both sea-level and d18Ob have been respectively reconstructed or measured. Crucially, Bates et al. note that "*The calibration windows for sea level are chosen as prolonged interstadial or stadial events, when sea level and temperature are at approximate equilibrium*" and that the method is not suitable during glacial inceptions and terminations – as we noted in original Lines 307-312. Unfortunately, the interglacials are too short, especially those from MIS 11 onwards, for ice sheets to reach approximate equilibrium with climate, as this would require tens of thousands of years of constant temperature (for example the Antarctic Ice Sheet: see Garbe et al., 2020). In contrast, deep ocean temperature responds to surface climate change at centennial time scales (Yang et al., 2011; Li et al., 2013). Hence, the ambient temperature signal in d18Ob might respond to global climatic changes over time scales of order 0.1 kyr, while the ice volume (d18Osw) signal likely responds over time scales of 10 kyr, potentially biasing reconstructions with this method during the rapid climate changes encountered through interglacials. Adding extra sites based on d18O can help improve the signal to noise ratio at our current 4 kyr temporal resolution, and can help by targeting a more geographically relevant region, but we would essentially be reinforcing an underlying bias during interglacials (which are often the periods of most interest). This same limitation also prevents us from justifying an increased temporal resolution even if there are sufficient data to do so from a statistical aspect. Consequently, these extra Southern Ocean sites could certainly benefit a 800 kyr coarse resolution synthesis, but we would not gain much useful information by attempting a higher resolution synthesis over a shorter period e.g. since MIS 11.

(ii)   We have reviewed ~130 benthic d18O records south of 40 degS in the Mulitza database. For our purposes, records need to cover several glacial cycles with sufficient temporal resolution, to identify calibration windows and confidently establish a transfer function; they also need to represent a suitable water mass (ideally lower CDW). We also require records to extend back to at least MIS 11, so that our results from MIS 11 to present are not impacted by a changing spatial distribution of records. Based on those selection criteria we have identified six sites, of which only one (ODP 1090) was included in Bates et al. 2014.

(iii)   We would need to bear in mind a synthesis with many d18O sites and only two Mg/Ca sites (assuming we exclude the North Atlantic sites), will be heavily biased towards one method.

(iv)   For the two Southern Ocean sites with Mg/Ca, calculation of BWT using both Mg/Ca and d18O would provide an interesting comparison.

Given that analysis of the six selected sites seems worthwhile, we also take this opportunity to develop the method used by Bates et al. 2014. This includes updated and additional sea-level markers, and use of 2$^{nd}$ or 3$^{rd}$-order polynomials instead of linear piece-wise transfer functions. The site selection and analysis of d18O are now described in detail in the methods Sections 3.1 and 3.3. Details of the sea-level markers and sites are included in the new Tables 1 and 2.

**(1.3) Additionally, the new compilation shows strong agreement to past DWT compilations - so I'm not entirely sure that is really adds much to the literature. I was really hoping it would get into how it might influence the ice sheets or modelling so that it did add something new.**

It is true that our original synthesis agreed closely with the Rohling et al. (2021) global mean DWT reconstruction, when their reconstruction has been resampled to match our 4 kyr resolution. This good agreement is generally maintained in the revised version (new Figs 5, 8; sections 4 and 5.1), except perhaps during glacial periods when we find less cooling (interpreted as their being less possibility to cool water that is already closer to its freezing point). Good agreement with Rohling et al was not an inevitable outcome, given the very different geographic scope (Southern Ocean CDW versus global deep water), and we view this as an interesting result rather than a weakness. We would also note the agreement is strong at the coarse 4 kyr resolution that we used, but as more data become available and a higher resolution synthesis is feasible, it is of course possible that periods with weaker agreement will emerge.

Regarding the influence on ice sheets or modelling, please see our response to Point 1.1 above.

**(1.4) I also find the inclusion of the NADW records (4 out of the total of 7) a little odd given the focus on the LCDW and evidence in the literature of the shut down of AMOC and therefore I would not expect the NADW to be a strong contributor to LCDW during the glacials. The authors came to this eventually in the paper, but only after putting it all together and didn't really discuss the implications on their records which have 4 records from NADW that may well swamp the LCDW datasets.  Although I realise the authors are interested in the interglacials for the ice sheet melting. But then didn't really come back to this and how the MIS5, 9 and 11 may have impacted the ice sheets.**

It is indeed unintuitive to include NADW (particularly from N Atlantic sites) - this decision was originally motivated by the use of the North Atlantic ODP 980 BWT anomaly directly as an ocean temperature boundary condition in Antarctic Ice Sheet modelling (Quiquet et al. 2018, Crotti et al. 2021). In our original version we already discussed this issue of whether NADW records should be used, and compared the records for CDW and NADW, which it seems are well correlated – there was actually better agreement between the NADW and CDW subsets of records, than between the d18O and Mg/Ca subsets (original Fig 4). However, now that we have analysed new Southern Ocean sites it is of course better to not include the North Atlantic sites. We have been clearer about this point in the revised Methods (Section 3.3 "Site selection").

We also compare our synthesis with site ODP 980 (Figs 8 & 9).

**(1.5) Many sections there was insufficient detail on the methods and the issues – they came back to many of these in the discussion at the end – but they should have been upfront. There really was insufficient detail around the methods used for the Mg/Ca and the d18O. I realise these are in the original papers – but you need to provide a summary here so the reader doesn't have to go back to the original papers.**

Both methods are now described in much more detail, in the Methods (Section 3).

**(1.6) I felt the oceanography background needed more information and detail on the CDW – and differentiate between the NADW, LCDW, UCDW and mCDW – it would have helped to have a 3D/Depth figure to show the links between these water masses.**

We have revised the oceanography background (Section 2) and added a new schematic Fig 1 summarising water masses in the Atlantic sector, based on slightly different versions of this schematic by Howe et al. (2016), Ferrari et al. (2014) and Matsumoto (2017). Talley (2013) produced a clear 3D figure of circulation in the Southern Ocean, we have referred to that in the Fig 1 caption (rather than copying it directly into this paper).

**(1.7) It was unclear to me how the authors plotted up Figure 4 when you can't look at BWT on both the NADW and the LCDW at the same time? How do you make plots like this? Also which cores had both benthic d18O and the Mg/Ca done on them to make the other X-Y plot.**

The original Fig 4 is no longer used in the revised version. We now compare the two proxies (Mg/Ca and d18O) in the new Fig 6, where it is hopefully clear that each point represents a 4 kyr time slice in one core.

**(1.8) The reason why there are not many DWT records using Mg/Ca for benthic foraminifera is that it is not a trivial thing to do. Firstly there are not always sufficient tests of the right species of benthic foraminifera. Secondly the method is time consuming and has some issues – some of which were outlined in the limitations. But one thing that has not been mentioned in the paper is the potential impact of dissolution and the Carbonate Compensation Depth - the LCDW of the Southern sits at or very close to the Carbonate Saturation Horizon which is around 3000 m and the CCD which sits around 4000 m- so not many cores actually preserve sufficient Carbonate organisms or the records can be compromised by period of dissolution due to the shoaling of the CCD - during the glacial cycle.**

Yes in our original manuscript the preservation issue was mentioned only very briefly (calcite dissolution) and not specifically linked to CCD, which is indeed an important limitation on where samples can be collected (particularly for reconstructions of AABW). This aspect has been addressed in the new methods (Section 3), where we also clarify that sample collection is not a trivial task [Lines 293-299].

**Reviewer #2**

**Chandler and Langebroek compiled 7 deep water records (3 benthic Mg/Ca, 1 ostracod Mg/Ca, and 3 benthic d18O) during the last 800 ka, aiming to understand the glacial-interglacial temperature variability of Circumpolar Deep Water. However, there are several problems with data selection, data interpretation, and presentation, I have to say that this study, at least in the current form, is not to the standard of the CP.**
**Here are my main concerns.**

**(2.1) The premise of using a compilation of BWT at selected sites to reflect CDW BWT change. Including NADW sites in the compilation is not appropriate because 1) NADW sites could be bathed in different water masses during glacials and interglacials due to circulation change, as already mentioned by the authors in Section 5.3, 2) shallow NADW sites (e.g. ODP 980) would have minimal influence on CDW that can potentially upwell underneath the Antarctic Ice shelves, due to the lower seawater density. On the contrary, Pacific sites at relatively deep depths, though downstream of CDW, may more reliably record the CDW temperature as those could be on the same isopycnal as CDW upwelling underneath the Antarctic Ice shelves. More benthic d18O records from the Pacific included in Bates et al (2014) thus might be included in the compilation as well. But ultimately, it is not most convincing to include sites not bathed in CDW to infer CDW changes. There may be more SO sites available if the time span of the compilation can be shortened.**

As we have noted in response to Reviewer #1 (Point 1.4 above), the use of NADW in the North Atlantic to reconstruct CDW in the Southern Ocean is certainly not intuitive, even though NADW does eventually mix with CDW. That decision was originally motivated by the use of N. Atlantic site ODP 980 directly as an ocean temperature boundary condition in some recent Antarctic Ice Sheet simulations (Quiquet et al., 2018; Crotti et al., 2022). It also reflected the lack of Southern Ocean sites for which temperatures had already been analysed.

In our revised synthesis we have abandoned the North Atlantic sites and instead have analysed additional d18O records from the Southern Ocean, including three in the Indian-Pacific sector (Table 2 and new Fig. 2). The methods and site selection are now described in far more detail than originally, in the new methods (Section 3).

**(2.2) Using a compiled CDW T to inform ice shelf melting triggered by CDW.**

**The authors set out to use a compiled CDW T record to infer potential ice shelf melting triggered by CDW during previous warm interglacials. It is noted that such an event can be triggered by ~1 degC warming in a short period (a year) in the modern ocean (e.g., Jenkins et al., 2018), which is a really small difference challenging for any given record reconstructed by any proxy to resolve with confidence, let alone a compilation smoothing multiple records. So the 0.1-0.5degC warming during previous interglacials mentioned in the abstract must be interpreted in the context of the uncertainty range, which is not reported but can be expected to be large enough to make the reported warming statistically insignificant.**

First we would like to clarify that "*The authors set out to use a compiled CDW T record to infer potential ice shelf melting triggered by CDW during previous warm interglacials*" was beyond our original stated intention of compiling a temperature synthesis (original Lines 63-64). We had not implied we would go on to use our synthesis to estimate changes in ice shelf melting, although the study is motivated by the need for an improved ocean temperature boundary condition in ice sheet models. Our objectives and scope are hopefully now clearer in the Introduction [Lines 70-77].

It is true that the strong sensitivity of basal melt to temperature changes of 1 degC or even less is a major challenge for using proxy records – or indeed any other existing method – for estimating basal melting prior to the start of modern observations. Although additional records might improve the spatial and temporal resolution of CDW temperature reconstructions, they are unlikely to confidently capture decadal or even centennial changes that are also important for ice shelf melt and grounding line migration (Jenkins et al., 2018). If proxy records are used as the basis for the ocean temperature boundary condition in an ice sheet model, they will only provide a long-term baseline, so that shorter time-scale variability would ideally be added as stochastic noise with characteristics informed by other methods (e.g., modern observations or numerical modelling). We have now noted this limitation at [Lines 531-532]. The problem of smoothing is also now discussed, in Section 5.2.

We have added uncertainty ranges to temperature anomalies quoted in the text.

**(2.3) Hydrographic settings of CDW.**

**As a manuscript on CDW, there are lots of inaccurate statements about CDW. While NADW is the heat source of the modified CDW, it is not the principal source water mass of CDW (Line 7). Deep waters from the Indian and Pacific sectors, as well as deep water formed around Antarctica, are also key sources of CDW (refer to Talley 2011 for detailed hydrography). Also, UCDW and LCDW are not defined properly. And modified CDW is often used in the context of melting under ice shelves, but it is not a counterpart of UCDW and LCDW.**

The oceanography background (Section 2) has been revised based on Talley (2013) and Carter et al. (2021), as well as other recent studies referred to in Section 2. Our usage of CDW follows its usage in those references.

**(2.4) Lack of details about their method.**

**how the BWT is calculated? How is the sampling done? Which types of uncertainties are included in the uncertainty envelope shown in Figure 4? These questions are not trivial. Details, such as choice of d18O-T equation, treatments of ice volume change, and local effects**

**on seawater d18O, are lacking for benthic d18O calculation. For the sampling method, no original data is shown, so one would have no way to assess if the compilation is potentially biased by one particular site.**

The method for calculating BWT from d18O is now described in a lot more detail in Section 3.1.2. Plots for individual sites are now included in Fig. 3.

**(2.5) The structure of the paper.**

**A reader is left with the impression that the work done here did not at all contribute to solving the research question raised in the introduction, i.e., the potential influence of CDW T on ice shelf melting during previous interglacials. The introduction sets the expectation of a reader very high and by finishing reading I felt a little disappointed. Perhaps, the authors could put something that can be achieved by this work in the introduction. Also, I think it would be better to provide some background on the limitations of the employed proxies in the introduction.**

Regarding the aims and scope of the paper in terms of influence on ice shelf melting, this was also raised by Reviewer #1. We have now added Section 5.4 and Fig. 10 on potential changes in thermal forcing in three different sectors of Antarctica, noting the important caveat that we reconstruct only CDW temperature and not the rate of its transport into ice shelf cavities.

We agree limitations were not dealt with in sufficient detail in our original manuscript (the reader was simply referred to original publications) and these are now described in more detail in the methods (Section 3).

References

Albrecht et al. (2020), Cryosphere, https://doi.org/10.5194/tc-14-599-2020.
Bates et al. (2014), QSR, https://doi.org/10.1016/j.quascirev.2014.01.020.
Carter et al. (2021), https://doi.org/10.1016/B978-0-12-819109-5.00003-7.
Crotti et al. (2022), Nat. Comms, https://doi.org/10.1038/s41467-022-32847-3.
Garbe et al. (2020), Nature, https://doi.org/10.1038/s41586-020-2727-5.
Ferrari et al. (2014), PNAS, https://doi.org/10.1073/pnas.1323922111.
Howe et al. (2016), Nat. Comms., https://doi.org/10.1038/ncomms11765.
Jenkins et al. (2018). Nat. Geos. https://doi.org/10.1038/s41561-018-0207-4.
Jouzel et al. (2007), Science, https://doi.org/10.1126/science.1141038.
Li et al. (2013), Clim. Dyn., https://doi.org/10.1007/s00382-012-1350-z.
Matsumoto (2017), PNAS, https://doi.org/10.1073/pnas.1701563114.
Mulitza et al. (2022), ESSD, https://doi.org/10.5194/essd-14-2553-2022.
Nürnberg et al. (2004), Geophys. Mono. Ser., https://doi.org/10.1029/151GM17.
Quiquet et al. (2018), GMD, https://doi.org/10.5194/gmd-11-5003-2018.
Reese et al. (2018), Cryosphere, https://doi.org/10.5194/tc-12-1969-2018.
Rohling et al. (2021), Sci Adv, https://doi.org/10.1126/sciadv.abf5326.
Sutter et al. (2019), Cryosphere, https://doi.org/10.5194/tc-13-2023-2019.
Talley (2013), Oceanography, https://doi.org/10.5670/oceanog.2013.07.
Tigchelaar et al (2018), EPSL, https://doi.org/10.1016/j.epsl.2018.05.004.
Yang et al. (2011), GRL, https://doi.org/10.1029/2011GL048076.

---

## Author Response (AR2)

17 July 2024

Dear Erin McClymont

Many thanks for your comments on our manuscript. We have addressed these below, with corresponding line numbers referring to the latest track-changes version.

Best wishes

Dave Chandler
* * *
**- line 77 "considered less reliable…" Can you clarify that you mean "less reliable as temperature proxies" rather than that the oxygen isotopes are less reliable in general?**

We have now written this as "deep water temperature proxy records derived from oxygen … are considered less reliable prior to the MPT" (Line 68).

**- Line 180 refers to "local hydrographic controls" perhaps influencing d18Osw but not at glacial interglacial timescales. But given the formation and transport processes of LCDW as described in section 2 (E.g. lines 105-110) is there a chance that mixing with Antarctic waters, which might been more strongly influenced by local impacts on temperature and salinity, could impact LCDW properties?**

Yes, if Antarctic water masses (particularly AABW) have a distinct oxygen isotopic signature gained e.g. from ice sheet freshwater fluxes, then this could influence LCDW $\delta^{18}O_{sw}$ as northbound AABW gradually mixes with southbound NADW/LCDW in the lower overturning cell. The glacial-interglacial *change* in LCDW $\delta^{18}O_{sw}$ is very similar to that of the global average (see estimates of $\Delta\delta^{18}O_{swLGM}$ at Line 211), suggesting this local influence is relatively small at our sites, on glacial time scales. However we have highlighted this as a potential avenue for future work, whether focusing on glacial cycle or shorter time scales, at Line 166.

**- Related to the previous comment, lines 111-120 there is discussion of the modification of LCDW before it reaches the ice front. In the discussion (line 694) the authors flag the caveat that they reconstruct temperature and not transport: but would they also expect a potential cooling during cross-shelf transport which would also mean their reconstructed temperatures lie towards the maximum expected?**

Cross-shelf cooling by mixing with other water masses does mean our reconstructed LCDW temperatures are likely an upper bound on water temperature reaching Antarctic grounding lines. However our synthesis reports a temperature *anomaly*, rather than in-situ temperature. The temperature anomaly at the grounding line should follow the open-ocean LCDW temperature anomaly but with potential further modifications by (1) changes in degree of mixing and (2) changes in temperature of

water masses with which LCDW is mixed. It's quite speculative to discuss in detail how that would modify temperature anomalies at grounding lines (they could be adjusted up or down, depending on changes in transport and mixing under past climates). In the text we now note our reconstruction does not account for additional influences of CDW transport rate or modification across the shelf (Line 509).

**- Line 392 describes the number of sites used. Two of the sites have two proxies for Tcdw being used, so are they being treated as four separate records (4 sites?). What would happen if only Mg/Ca was used for the two sites where it was available, along with d18O where it was not? Although the potential bias between proxies or regions was explored, the potential bias caused by two records coming from the same site was not.**

At sites with both proxies, the two corresponding records are treated as independent estimates in the statistical analysis used to construct means and confidence intervals - so yes, four records from two sites. This is reasonable if most variance in each time slice derives from methodological errors, but not if most variance derives from geographical variability. From Figs 6 & 7 we would suggest it is methodological errors contributing the bulk of the variance, e.g. given there is a stronger match between the Pacific and Atlantic sector sites, than between temperatures reconstructed at the same site with different proxies. See discussion at Lines 405-415. However, there is potential for a slight underestimate of confidence interval widths if the two records from one site are not fully independent. We have added a note of this in the methods at Lines 358.

Use of Mg/Ca where possible, otherwise $\delta^{18}O$:
At ODP 1094 we would end up with a mix of the two proxies, while at ODP 1123 we would almost exclusively use Mg/Ca (very few 1123 time slices have $\delta^{18}O$ and no Mg/Ca). The small differences this makes to the mean at each time slice (Fig R1 below) are not statistically significant given the uncertainty – particularly considering reducing $N$ by 2 sites considerably widens the error bars when $N$ is already small. These are interesting questions but really need more sites to answer with any confidence. We have noted in the priorities for future work, that application of multiple proxies to the same core samples will be beneficial to quantify biases independently of other sources of variance (Line 578).

[Figure]

Fig R1: Comparison between DWT estimated using (i) our original synthesis (black line) and (ii) using Mg/Ca where possible but otherwise $\delta^{18}O$ (blue line).

**- Line 415 the authors note the Mg/Ca records tend to give cooler glacials and warmer interglacials, so a wider range than suggested by their d18O reconstruction. But the stack is dominated by d18O, so should we treat the stack as a conservative prediction of Tcdw? (Or might this observation explain the quadratic nature of the relationship with other archives due to systematic bias or underestimation of the lowest temperatures?)**

If we assume that Mg/Ca is the more reliable of the two proxies, then yes the relatively weaker glacial-interglacial response of $\delta^{18}O$ temperatures would imply our stacked temperature changes are conservative, given the dominance of $\delta^{18}O$ in the stack. However, if it turned out $\delta^{18}O$ was more reliable, then including Mg/Ca would conversely imply that our stack tends to overestimate temperature changes. We haven't commented on which might be the most reliable of the two proxies as we cannot objectively evaluate that with the information we have. A third proxy would be helpful in this respect as now clarified in the priorities for future reconstructions (Line 575).

Regarding the quadratic relationship:
If $\delta^{18}O$ yields less glacial cooling but also less interglacial warming, relative to Mg/Ca, we would still expect linear relationships between reconstructed $T_{CDW}$ and the other temperature indicators, assuming the 'real' relationship with $T_{CDW}$ is linear. Instead we find a quadratic fit, as the $T_{CDW}$ gradient reduces at cooler temperatures. This most likely reflects the already cold temperature of LCDW, which limits the potential for cooling under peak glacial conditions. For example, glacial cooling of AIS surface air temperature is unlimited (in a practical sense) while $T_{CDW}$ cannot cool below its freezing point. See discussion starting at Line 423.

**- Figure 5: I can't see the vertical shading described in the text, whether printed or on screen.**

We have now darkened the shading so it's easier to see.